

# Measurement report:Assessing the Impacts of Emission Uncertainty on Aerosol Optical Properties and Radiative Forcing from Biomass Burning in Peninsular Southeast Asia

Yinbao Jin[1], Yiming Liu*[1], Xiao Lu[1], Xiaoyang Chen[2], Ao Shen[1], Haofan Wang[1], Yinping Cui[1], Yifei Xu[1], Siting Li[1], Jian Liu[1], Ming Zhang[4], Yingying Ma[3], Qi Fan*[1,5,6]

[1]School of Atmospheric Sciences, Sun Yat-Sen University, Zhuhai 519082, China

[2] Institute of Tropical and Marine Meteorology, China Meteorological Administration, Guangzhou, 510000, China

[3] State Key Laboratory of Information Engineering in Surveying, Mapping and Remote-sensing, Wuhan University, Wuhan 430079, China

[4]Hubei Key Laboratory of Critical Zone Evolution, School of Geography and Information Engineering, China University of Geosciences, Wuhan 430074, China

[5]Southern Marine Science and Engineering Guangdong Laboratory, Zhuhai 519082, China

[6]Guangdong Province Key Laboratory for Climate Change and Natural Disaster Studies, School of Atmospheric Sciences, Sun Yat-sen University, Guangzhou 510275, China

*Correspondence to*: Qi Fan (eesfq@mail.sysu.edu.cn) and Yiming Liu (liuym88@mail.sysu.edu.cn)

**Abstract.**

Despite significant advancements in improving the dataset for biomass burning (BB) emissions over the past few decades, uncertainties persist in BB aerosol emissions, impeding the accurate assessment of simulated aerosol optical properties (AOPs) and direct radiative forcing (DRF) during wildfire events in global and regional models. This study assessed AOPs (including aerosol optical depth (AOD), aerosol absorption optical depth (AAOD), and aerosol extinction coefficients (AEC)) and DRF using eight independent BB emission inventories applied to the WRF-Chem model during the BB period (March 2019) in Peninsular Southeast Asia (PSEA), where the eight BB emission inventories were the Global Fire Emissions Database version 4.1s (GFED), Fire INventory from NCAR version 1.5 (FINN1.5), the Fire Inventory from NCAR version 2.5 MOS (MODIS fire detections, FINN2.5 MOS), the Fire Inventory from NCAR version 2.5 MOSVIS (MODIS+VIIRS fire detections, FINN2.5 MOSVIS), Global Fire Assimilation System version 1.2s (GFAS), Fire Energetics and Emissions Research version 1.0 (FEER), Quick Fire Emissions Dataset version 2.5 release 1 (QFED), and Integrated Monitoring and Modelling System for Wildland FIRES Project version 2.0 (IS4FIRES), respectively. The results show that in the PSEA region, organic carbon (OC) emissions in the eight BB emission inventories differ by a factor of about 9 (0.295-2.533 Tg/M), with $1.09 \pm 0.83$ Tg/M and a coefficient of variation (CV) of 76%. High-concentration OC emissions occurred primarily in savanna and agricultural fires. The OC emissions from the GFED and GFAS are significantly lower than the other inventories. The OC emissions in FINN2.5 VISMOS are approximately twice as high as those in FINN1.5. Sensitivity analysis of AOD simulated by WRF-Chem to different BB emission datasets indicated that the FINN scenarios (v1.5 and 2.5) significantly overestimate AOD compared to observation (VIIRS), while the other inventories underestimate AOD in the high AOD (HAOD, AOD>1) regions





range from 97-110°E, 15-22.5°N. Among the eight schemes, IS4FIRES and FINN1.5 performed better in terms of AOD
simulation consistency and bias in the HAOD region when compared to AERONET sites. The AAOD in WRF-Chem during
the PSEA wildfire period was assessed using satellite observations (TROPOMI) and AERONET data, and it was found that
the AAOD simulated with different BB schemes did not perform as well as the AOD. The significant overestimation of AAOD
by FINN (v1.5 and 2.5), FEER, and IS4FIRES schemes in the HAOD region, with the largest overestimation for FINN2.5
MOSVIS. FINN1.5 schemes performed better in representing AAOD at AERONET sites within the HAOD region. The
simulated AOD and AAOD from FINN2.5 MOSVIS always show the best correlation with the observations. AEC simulated
by WRF-Chem with all the eight BB schemes trends were consistent with CALIPSO in the vertical direction (0.5 km to 4 km),
demonstrating the efficacy of the smoke plume rise model used in WRF-Chem to simulate smoke plume heights. However,
the FINN (v1.5 and 2.5) schemes overestimated AEC, while the other schemes underestimated it. In the HAOD region, BB
aerosols exhibited a daytime shortwave radiative forcing of $-32.60\pm24.50$ W/m$^2$ at the surface, positive forcing ($1.70\pm1.40$
W/m$^2$) in the atmosphere, and negative forcing ($-30.89\pm23.6$ W/m$^2$) at the top of the atmosphere. Based on the analysis,
FINN1.5 and IS4FIRES are recommended for accurately assessing the impact of BB on air quality and climate in the PSEA
region.

## 1 Introduction

Peninsular Southeast Asia (PSEA), including Vietnam, Thailand, Myanmar, Cambodia, and Laos, is one of the major biomass
burning (BB) emission source areas in the world (Yadav et al., 2017). Due to widespread forest fires and agro-residue burning,
extensive BB activities occur over PSEA, especially during the dry season (BB usually peaks in March) (Reddington et al.,
2021)  and release large amounts of aerosols and trace gases (including organic carbon (OC), black carbon (BC), particulate
matter (PM), nitrogen oxides (NO$_x$), and volatile organic compounds (VOC)) into the air, thus leading to significant impacts
on atmospheric composition, radiative budget, and human health (Reid et al., 2013). Therefore, it is crucial to understand the
BB emission inventories, as well as the behavior of aerosols, and accurately model their properties, to assess their impact on
air quality and climate change in the PSEA region.
Numerous studies have been conducted to assess the effects of BB emissions on aerosol optical properties (AOPs), such as
aerosol optical depth (AOD), absorbing aerosol optical depth (AAOD), and aerosol extinction coefficient (AEC), as well as
direct radiative forcing (DRF) in the PSEA region (Zhu et al., 2017; Lin et al., 2014; Dong and Fu, 2015a).  However, most of
these studies have relied on only one single BB emission inventory without comparing different inventories, leading to large
uncertainties in assessing the impact of BB aerosols. Due to the challenges in directly measuring BB emissions, various global
fire emissions inventories have been developed based on satellite observations in the past decades (Ichoku and Ellison, 2014;
Wiedinmyer et al., 2023; Wiedinmyer et al., 2011). These inventories use different empirical methods and underlying data to
represent gas and aerosol emissions from fires, resulting in inherent uncertainties (Carter et al., 2020).



These uncertainties arising from different BB emissions often manifest as regional variations and inconsistencies with
observations when integrated into models (Liu et al., 2020). Addressing these uncertainties is crucial for refining climate
models and providing more accurate projections of future climate change. For example, Pan et al. (2020) compared six BB
aerosol emission datasets from 2008 globally as well as from 14 regions, and the total global emissions from these BB emission
datasets differed by a factor of 3.8. Sensitivity analysis of AOD simulated by Goddard Earth Observing System-Chemistry
(GEOS-Chem) to different BB emission datasets during the peak BB period in each region and at most AERONET sites in
each region found that Quick Fire Emissions Dataset version 2.4 (QFED2.4) produced the highest AOD values, closest to
observations, followed closely by Fire Energetics and Emissions Research version 1.0 (FEER1.0). In the North American
region, the GEOS-Chem incorporating four different BB emission inventories and remote-sensing data analysis during wildfire
periods indicated a 4 to 7-fold difference in BB aerosol emissions. Simulations driven by Global Fire Emissions Database
version 4s (GFED4s) and Global Fire Assimilation System version 1.2 (GFAS1.2) provide better agreement with surface
measurements of organic aerosol and BC mass concentrations, BC observations at higher altitudes, and Moderate Resolution
Imaging Spectroradiometer (MODIS) observations of AOD (Carter et al., 2020). To explore the uncertainty of BB emissions
in the tropics, GFED V3, Fire INventory from NCAR version 1 (FINN1.0), and GFAS1 were used to evaluate Global Model
of Aerosol Processes (GLOMAP) model simulations of AOD in South America, Africa, and Southeast Asia showing that the
model underestimates AOD for all emission datasets (Reddington et al., 2016). In the North Sub-Saharan Africa BB region,
Zhang et al. (2014) found a 12-fold difference in estimates of total smoke emissions and an even larger difference (up to 33-
fold) in WRF-Chem simulated smoke-related variables and radiative effects. Recent studies have shown that during the PSEA
march BB peak, only the FINN2.5 captures the feature, which is not seen in GFED and as pronounced in other inventories
(Wiedinmyer et al., 2023). Despite substantial research efforts, accurately representing BB aerosols in models remains a
challenge. In summary, compared to the differences between global BB emission inventories, regional differences may be
larger, especially in the PSEA region, where the satellite inversions of BB contain a large fraction of uncertainty due to high
cloud cover (Dong and Fu, 2015b). Significant differences exist in AOPs and radiative forcing simulated by different emission
inventories in the high BB emission region within a single model (Carter et al., 2020; Zhang et al., 2014). To reduce
uncertainties, it is necessary to compare the differences between commonly used BB emission inventories and evaluate the
model simulations of AOPs and radiative effects for the PSEA region.
In March 2019, the National Aeronautics and Space Administration (NASA) used remote sensing data from Visible Infrared
Imaging Radiometer Suite (VIIRS) to discover hundreds of fires burning in the PSEA region (Jenner, Mar 18, 2019). Therefore,
this study aims to analyze how emission uncertainties or differences from different BB inventories affect the spatial and
temporal distribution of aerosols and their radiative effects in the PSEA region. Section 2 describes the model configuration,
experimental design, and data sources. Section 3 presents a comparison of eight emission inventories in March 2019 and the
results of simulating AOPs and DRF. Discussions are provided in Section 4, and the study concludes with a summary in
Section 5.



## 2 Data and Methods

### 2.1 Model Description and Configuration

#### 2.1.1 WRF-Chem

The simulations were conducted using version 3.9.1.1 of the WRF-Chem online-coupled meteorology and chemistry model (Grell et al., 2005). A single domain (Figure 1) was employed, mainly focusing on the PSEA region (red line, including Vietnam, Thailand, Myanmar, Cambodia, and Laos,) and studying BB from February 26th to March 31st, 2019. The initial 3 days of the model simulation were used as a spin-up period. The model consisted of 27 vertical layers and one nested horizontal resolution of 27 x 27 km. The selected physical configurations included the Morrison double-moment microphysics scheme (Morrison et al., 2005), the Rapid Radiation Transfer Model (RRTMG) longwave and shortwave radiation schemes (Iacono et al., 2008), the Mellor-Yamada-Janjic (MYJ) planetary boundary layer scheme (Mellor and Yamada, 1982; Janjić, 1990), the Eta similarity surface Layer scheme (Monin and Obukhov, 1954), the Noah Land Surface Model land surface scheme (Niu et al., 2011) and the Grell 3D cumulus parameterization scheme (Grell and Dévényi, 2002). The Model for Ozone and Related chemical Tracers (MOZART) trace gas chemistry with the Model for Simulating Aerosol Interactions and Chemistry (MOSAIC with 4 bins) aerosol scheme with the Kinetic Preprocessor (KPP) library is used in the model (Emmons et al., 2010). In this study, MOSAIC uses a sectional approach to represent aerosol size distributions with four discrete size bins with glyoxal uptake into aqueous aerosols to form secondary organic aerosol (SOA) in the PSEA region by WRF-Chem, which is capable of simulating all major aerosol components, including nitrates ($NO_3^-$), sulfates ($SO_4^{-2}$), ammonium ($NH_4^+$), BC, primary organic aerosols, and other inorganic aerosols, with high efficiency and accuracy for use in air quality and regional/global aerosol modeling (Zhang et al., 2018). The Community Atmosphere Model with Chemistry (CAM-chem) simulation outputs (Emmons et al., 2020; Buchholz et al., 2019) are used as chemical lateral boundary and initial conditions for WRF-Chem (https://rda.ucar.edu/datasets/ds313.7/, last access: 11 May 2023). The product simulated by CAM-chem has a horizontal resolution of 0.9 degrees by 1.25 degrees and 56 vertical levels in the vertical direction. Meteorological initial and boundary conditions were obtained from the National Centers for Environmental Prediction Final Analysis data with a 1° x 1° horizontal resolution.

WRF-Chem employs Mie theory to perform calculations of AOPs using MOSAIC size distributions and the complex refractive indices associated with each MOSAIC chemical constituent. Specifically, it simulates AOPs (such as AEC, single scattering albedo (SSA), and asymmetry factor for scattering) distributed in four different bands: 300, 400, 600, and 1000 nm. This study used the Ångström power law (Ångström, 1929; Martınez-Lozano et al., 1998) to derive the model at 550 nm for AOD, and the detailed calculation procedure follows Kumar et al. (2014) and Saide et al. (2013). In addition, the aerosol direct radiative feedback was coupled with the RRTMG for both shortwave (SW) and longwave (LW) radiation as implemented by Zhao et al. (2010). A detailed description of the computation of AOPs and DRF in WRF-Chem has been given by Fast et al. (2006), Zhao et al. (2011), and Lin et al. (2014).



### 2.1.2 Anthropogenic and Biogenic Emissions

The latest version of the global anthropogenic emissions inventory, the monthly Emissions Database for Global Atmospheric Research (EDGAR) v5.0, was published by Marvin (2022) on February 17, 2022. It provides global air pollutant emissions for the year 2015 at a resolution of 0.1°×0.1°. These emissions were speciated for the MOZART chemical mechanism and can be accessed at https://zenodo.org/record/6130621 (last accessed on 11 May 2023). Biogenic emissions were calculated online within the model using the Model of Emissions of Gases and Aerosols from Nature (MEGAN) inventory developed by Guenther et al. (2012).

### 2.2 BB Emission Inventories

There are two primary approaches to estimating BB emission inventories: "bottom-up" and "top-down" methods (Archer-Nicholls et al., 2015). The "bottom-up" approach involves estimating emissions per species by multiplying emission factors (EF) with estimates of the biomass burned (Yevich and Logan, 2003). The latter, the "top-down" approach, bypasses the largely uncertain fuel consumption estimation step by estimating emission fluxes directly from fire radiative power (FRP) (Ichoku and Ellison, 2014). The "top-down" approach commonly utilizes AOD retrieved from satellite remote sensing to constrain aerosol emissions from wildfires (Huneeus et al., 2012). This study evaluates the performance of the WRF-Chem using eight different BB emission inventories to simulate wildfires in the PSEA region during March 2019. These emission inventories include the Global Fire Emissions Database version 4.1s (GFED), Fire INventory from NCAR version 1.5 (FINN1.5), the Fire Inventory from NCAR version 2.5 MOS (MODIS fire detections, FINN2.5 MOS), the Fire Inventory from NCAR version 2.5 MOSVIS (MODIS+VIIRS fire detections, FINN2.5 MOSVIS), Global Fire Assimilation System version 1.2s (GFAS), Fire Energetics and Emissions Research version 1.0 (FEER), Quick Fire Emissions Dataset version 2.5 release 1 (QFED), and Integrated Monitoring and Modelling System for Wildland FIRES Project version 2.0 (IS4FIRES). Table 1 provides a detailed comparison of their spatial and temporal resolution, the main references for the EF, the satellite data sources, Non-methane hydrocarbons (NMHCs), oxygen volatile organic compounds (OVOCs), gas, and aerosols in the inventory. NMHCs refer to organic compounds containing only C and H besides methane ($CH_4$), such as alkanes, alkenes, alkynes, etc. OVOCs contain C, H, and O compounds, e.g., alcohols, aldehydes, ketones, etc. NMHCs and OVOCs combined constitute nearly all of the non-methane volatile organic compounds (NMVOCs) emitted by wildfires (Akagi et al., 2011).

### 2.2.1 GFED (v4.1s)

The GFED4.1s datasets provide the area burned, dry matter (DM), and EF from global fires. It has a spatial resolution of 0.25° x 0.25° and can be accessed at https://daac.ornl.gov/get_data/ (last accessed on 11 May 2023). This dataset includes fractional contributions from different fire types and offers daily or 3-hourly data to scale monthly emissions to a higher temporal resolution. GFED4.1s is an enhanced version of the GFED4 dataset, incorporating small fire inputs to enhance the accuracy and completeness of emission estimates (Randerson et al., 2017). It covers the period from June 1997 to 2022 and includes a



wide range of emission species such as carbon (C), DM, carbon dioxide ($CO_2$), carbon monoxide (CO), methane ($CH_4$),
hydrogen ($H_2$), nitrous oxide ($N_2O$), $NO_x$, NMHCs, OVOCs, OC, BC, PM less than 2.5 microns in diameter ($PM_{2.5}$), total PM
(TPM), and sulfur dioxide ($SO_2$). The raw GFED emission data (0.25°x 0.25°) were first re-gridded to the required spatial
resolution for the WRF-Chem domains using the Earth System Modeling Framework (EMSF) program in Figure 2, followed
by supplementing the GFED emission species (Table S1) to meet the MOZART-MOSAIC scheme based on the study by
Akagi et al. (2011) and Heil A. (2020). The construction of the final emission inventory included incorporating the mean
fraction and fire size of the four vegetation types (grassland, extratropical forest, savanna, tropical forest) from FINN1.5. This
incorporation enables WRF-Chem to calculate the smoke plume rise (Freitas et al., 2007; 2010).

**2.2.2 FINN (v1.5, v2.5 MOS, and v2.5 MOSVIS)**

The emissions estimation of FINN (v1.5 and 2.5 ) are based on the framework described by Wiedinmyer et al. (2011) and
Wiedinmyer et al. (2023), which utilizes two types of satellite observations: (1) MODIS fire detections and (2) active fire
detections from both MODIS and VIIRS. It provides global daily estimates of BB emissions for important gases and aerosols,
along with comprehensive specifications of total VOC emissions for three commonly used chemical mechanisms (MOZART-
T1, SAPRC99, and GEOS-Chem) in regional and global chemical transport models (https://www.acom.ucar.edu/Data/fire/,
last accessed on 11 May 2023). Since its release, FINN has been widely utilized by researchers to assess air quality during
wildfire events (Lin et al., 2014; Vongruang et al., 2017; Pan et al., 2020). The latest version, FINN v2.5, was introduced in
2022 and incorporates an updated algorithm for determining fire size by aggregating adjacent fire detections. Compared to
FINN1.5, FINN2.5 incorporates significant improvements in input data and processing methods for detecting fire activity,
characterizing annual land use/land cover and vegetation density, estimating burned area, and applying fuel loads across
different global regions (Wiedinmyer et al., 2023). In this study, FINN1.5 and FINN2.5 MOS (MODIS-only fire detections),
and FINN2.5 MOSVIS (MODIS+VIIRS fire detections) were used. Detailed information on emission species and factors can
be found in Tables S2 and S3.

**2.2.3 GFAS (v1.2)**

The GFAS provides data outputs that encompass spatially gridded FRP, DM burning, and BB emissions for numerous chemical,
greenhouse gas, and aerosol species (Andela et al., 2013). These data are globally available from 2003 to the present, with a
regular latitude and longitude grid resolution of 0.1° x 0.1° (https://ads.atmosphere.copernicus.eu/cdsapp#!/dataset/cams-
global-fire-emissions-gfas, last accessed on 11 May 2023). The latest version, GFAS 1.2, includes injection height daily data
(mean altitude of maximum injection and altitude of plume top), which are obtained from the plume rise model and IS4FIRES.
To ensure BB data quality, quality control procedures were applied to the MODIS data. In Figure 2, it is illustrated that GFAS
1.2 data put into the WRF-Chem process, where the missing emission species (Table S4) required for the MOZART-MOSAIC
scheme are added by Jose et al. (2017), Andreae and Merlet (2001), and Andreae (2019) method. Additionally, the mean





fraction and fire size of the four vegetation types were obtained from FINN1.5, and the 3-hour time allocation from GFED4.1s
was utilized for the GFAS scheme.

**2.2.4 FEER (v1.0-G1.2)**

In 2005, a new algorithm was developed by Ichoku and Kaufman (2005) to calculate BB emissions directly from FRP
measurements (https://feer.gsfc.nasa.gov/data/emissions/, last accessed on 11 May 2023). This approach aimed to overcome
the delays and uncertainties associated with other variables previously used. Subsequently, their work resulted in the release
of the FEER Ce v1.0 product, a global BB inventory with a resolution of 0.1° x 0.1°. In this study, the FEERv1.0-G1.2 product
utilizes the GFASv1.2 FRP dataset to provide daily data from 2003 to the present at a spatial resolution of 0.1° x 0.1°. It
includes species such as CO, $SO_2$, $NH_3$, $NO_2$, OC, BC, $PM_{2.5}$, NMHCs, among others. Notably, the GFASv1.2 dataset has also
been incorporated to ensure compatibility with the MOZART-MOSAIC scheme, as depicted in Table S5.

**2.2.5 QFED (v2.5r1)**

QFED emissions are estimated using the FRP method and draw on the cloud correction technique developed in the GFAS.
However, QFED employs a more sophisticated approach for non-observed land areas, such as those obscured by clouds (Koster
et al., 2015). Fire locations and FRPs are derived from MODIS Level 2 fire products (MOD14 and MYD14) and MODIS
geolocation products (MOD03 and MYD03). QFEDv2.5r1, covering the period from 2000 to 2023, provides daily average
emissions at a horizontal spatial resolution of 0.1° x 0.1°, encompassing information on OC, BC, $SO_2$, CO, $PM_{2.5}$, and other
species. It can be accessed from https://portal.nccs.nasa.gov/datashare/iesa/aerosol/emissions/QFED/v2.5r1/ (last accessed on
11 May 2023). Figure 2 shows the detailed process of QFEDv2.5r1 to ensure consistency with the MOZART-MOSAIC
program. Table S5 illustrates the addition of missing data.

**2.2.6 IS4FIRES (v2.0)**

IS4FIRES is based on a reanalysis of FRP data obtained from the MODIS on the Aqua and Terra satellites. The dataset covers
the period from 2000 to the present (Sofiev et al., 2009). IS4FIRESv2 emissions are global, with a spatial resolution of 0.1° x
0.1°, provided every 3 hours, and represented in five stacked vertical layers (http://silam.fmi.fi/thredds/catalog/i4f20emis-
arch/catalog.html, last accessed on 11 May 2023)  (Soares et al., 2015). It distinguishes between seven vegetation classes:
boreal, temperate, tropical forests, residual crops, grasses, shrubs, and peat. The linear relationship between FRP and PM is
based on the IS4FIRESv1 EF but scaled to vegetation class types using the BB EF described in Akagi et al. (2011). Additional
IS4FIRES emission species according to Jose et al. (2017), Andreae and Merlet (2001) and  Andreae (2019), Baró et al. (2021),
and Wiedinmyer et al. (2011) meet the WRF-Chem selected MOZART-MOSAIC scheme (Table S5). It is noteworthy that its
time allocation is processed using the self-contained 3 hours (Figure 2).





**2.3 Observations and Reanalysis Data**

**2.3.1 Satellite observations**

Remote sensing satellite observation is widely utilized to evaluate AOPs, as it offers several advantages (Palacios-Peña et al., 2018), including non-interference with observed samples, sensitivity to various properties, particularly AOPs relevant to wildfires, and the ability to provide different types of data products such as points, columns, or profiles (Reid et al., 2013). To assess the AOD of European wildfires simulated by WRF-Chem, Palacios-Peña et al. (2018) compared products from different satellite inversions of AOD and selected the best product for model evaluation. Following a similar research approach, we chose the following satellite products: MODIS, VIIRS, and Himawari-8. In addition, Cloud-Aerosol Lidar and Infrared Pathfinder Satellite Observation (CALIPSO) satellites were selected to evaluate AEC simulated by WRF-Chem with BB emissions. Detailed descriptions of various satellite parameters and algorithms can be found in a previous study (Ma et al., 2021).

For a comprehensive understanding of absorbing aerosols emitted by global/regional wildfires, the Tropospheric Monitoring Instrument (TROPOMI) on the Sentinel-5 Precursor (S5P) satellite, launched on October 13, 2017, was employed to assess AAOD (Torres et al., 2020; Filonchyk et al., 2022). TROPOMI is a high spectral resolution spectrometer that covers the ultraviolet (UV) to shortwave infrared regions in eight spectral windows, offering enhanced capabilities for atmospheric monitoring compared to OMI satellites (Veefkind et al., 2012). Operating in a push-broom configuration, TROPOMI provides a wide swath width of approximately 2600 km over the Earth's surface. The instrument boasts higher spatial resolution, wider observation range, increased sensitivity and accuracy, more measurement parameters, and higher temporal resolution, making it an advanced tool for atmospheric monitoring. The TROPOMI aerosol algorithm (TropOMAER), employed for atmospheric observations, uses observations at two near-UV wavelengths to calculate UV Aerosol Index (UVAI) and retrieve total column AAOD and SSA (Torres et al., 2020). The AOD retrieved using TropOMAER inversion on land exhibits a root-mean-square error (RMSE) comparable to the OMI retrieval (maximum 0.1 or 30%). The RMSE of AOD over water may be two times larger, while the RMSE of AAOD is estimated to be approximately 0.01 (Torres et al., 2020). For this study, the TropOMAER L2 product (https://search.earthdata.nasa.gov/, last accessed on 11 May 2023) with a spatial resolution of 7.5 km x 3 km was selected. The WRF-Chem simulated AAOD at 500 nm was derived based on the method proposed by Hu et al. (2016), utilizing SSA (500 nm) from TROPOMI and Equation (1), where λ represents the wavelength. The uncertainty in SSA is approximately 0.03 (Dubovik and King, 2000)

$$AAOD(\lambda) = [1\text{-}SSA(\lambda)] \times AOD(\lambda) \qquad (1)$$



**2.3.2 In-situ observations**

To assess the effect of AOPs during wildfires, Baro et al. (2017) and Lin et al. (2014) first validated the meteorological field and pollutants simulated by WRF-Chem. Therefore, in this study, the FINN 1.5 scheme (the most common scheme used by WRF-Chem) was selected for validation of the model output for meteorological parameters and pollutants. The selected meteorological parameters include 2 m temperature (T2), 2 m relative humidity (RH2), and 10 m wind speed (WS10). These data were obtained from the data-sharing website (https://rp5.ru/, last accessed on 11 May 2023) and their global weather station identifications can be found in Table S6. The $PM_{2.5}$ data used to assess the stability of the model were collected from multiple publicly available website datasets from China (https://quotsoft.net/air/, last accessed on 11 May 2023), Thailand (http://air4thai.pcd.go.th/webV2/history/, last accessed on 11 May 2023), and global public datasets (https://aqicn.org/data-platform/covid19/, last accessed on 11 May 2023), and their locations are shown in Table S7.

The AERONET (AErosol RObotic NETwork) project is a collaboration between NASA and PHOTONS (PHOtométrie pour le Traitement Opérationnel de Normalisation Satellitaire; Univ. of Lille 1, CNES, and CNRS-INSU), establishes a collaborative network involving ground-based remotely sensed aerosol networks. This project has been in existence for over 25 years and provides a long-term, continuous, and easily accessible public-domain database for aerosol research, including the optical, microphysical, and radiometric properties of aerosols. AOD and AAOD measurements from AERONET are based on multiple wavelength bands, including visible and near-infrared spectra. Common band ranges include 340 nm, 380 nm, 440 nm, 500 nm, 675 nm, 870 nm, etc. AOD and AAOD data are classified into three levels based on data quality: level 1.0 (unscreened), level 1.5 (cloud shielding and quality control), and level 2.0 (quality assurance). For this study, data at the 2.0 level were used, indicating that the data underwent cloud screening and quality assurance following the detailed procedures outlined by Smirnov et al. (2000). In the absence of cloud contamination, the uncertainty in AOD was estimated to be 0.01 to 0.02, depending on wavelength. AAOD was calculated using Equation (1).

**2.3.3 ERA5 Reanalysis data**

European Centre for Medium-Range Weather Forecasts (ECMWF) Reanalysis v5 (ERA5) is a global meteorological reanalysis dataset developed and maintained by the ECMWF (Hersbach et al., 2018). The ERA5 dataset is based on global observational data, satellite remote sensing data, and numerical model forecast data. It uses advanced data assimilation techniques to fuse data from these different sources to produce consistent and high-quality global meteorological reanalysis data. Hourly data are available from 1979 up to the current time, and ERA5 data have a spatial resolution of 0.25° x 0.25° (about 25 km) at the horizontal level. In this paper, the effect of ERA5 950 hpa wind on BB aerosols is analyzed.

**2.4 Methodology**

In order to assess AOD, AAOD, AEC, and DRF using WRF-Chem with different BB inventories, apart from the FINN schemes, other emissions inventories are re-gridded and time-allocated, as shown in Figure 2. Subsequently, species are supplemented



according to the gas-phase chemistry and aerosol scheme (MOZART-MOSAIC) employed by WRF-Chem. It is worth noting
that all scenarios utilized fire size and vegetation type proportion data from FINN1.5 to calculate smoke plume rise. The
performance of WRF-Chem model simulations against measurements is evaluated using statistical metrics (Wu et al., 2019)
including the mean bias (MB), RMSE, Correlation coefficient (R), and the index of agreement (IOA) in Table S8. This research
further investigated DRF over PSEA during the study period. Zhao et al. (2013) and Lin et al. (2014) were referenced for the
treatment of BB aerosol radiative forcing, as shown in the following equations.

$$\mathbf{DRF} = \left(F_i^{\downarrow} - F_i^{\uparrow}\right) - \left(F_{no-fire}^{\downarrow} - F_{no-fire}^{\uparrow}\right) \qquad (2)$$

where $F^{\uparrow}$ and $F^{\downarrow}$ indicate the aerosol upward radiation flux and the aerosol downward radiation flux, respectively. $i$ indicates
that WRF-chem is added to the different BB emission inventories, and *no-fire* denoted scene without BB inventory applied.

## 3 Result

### 3.1 Inter-comparison of Eight BB Inventories.

Several studies have utilized OC as a measurable metric to compare variations among multiple BB inventories (Reddington et
al., 2016; Carter et al., 2020). This is because OC is a major component in smoke particles from fresh BB, with mass fractions
ranging from 37% to 67% depending on the fuel type (Pan et al., 2020). Figure 3 presents the spatial distribution characteristics
of OC for the eight BB datasets in the study region, along with the total OC emissions in the PSEA region during March 2019.
The highest OC emissions across all datasets are observed in the northern regions of Laos, Cambodia, and Thailand, as well
as in eastern and western Myanmar and southern Bangladesh. Lower emissions are observed in the central regions of Myanmar
and Thailand, northern Vietnam, and southern regions of China. Similar spatial distribution characteristics of OC emissions in
the PSEA region during March have also been reported by Pan et al. (2020) and Reddington et al. (2021). These emissions
mainly originate from shrubland, evergreen broadleaf, mixed shrubland/grassland, and dryland cropland, as classified by the
WRF-Chem land use data in the PSEA (Figure S1). The eight BB emissions, ranked based on their total OC emissions (PSEA)
in descending order, are FINN2.5 MOSVIS (2.533 Tg/M), FINN2.5 MOS (2.002 Tg/M), QFED (1.303 Tg/M), FINN1.5 (1.214
Tg/M), IS4FIRES (0.604 Tg/M), FEER (0.462 Tg/M), GFAS (0.296 Tg/M), and GFED (0.295 Tg/M). The highest OC
emission in the dataset is exhibited by FINN2.5 MOSVIS, which can be attributed to the use of updated burned area data and
the inclusion of fire information from VIIRS, capturing a larger number of small-scale fires (Wiedinmyer et al., 2023). The
lowest OC emissions are provided by GFED, which may have underestimated DM and agricultural fire EF (OC, EF=2.3 g/kg),
and GFAS, which only underestimated DM. The overall mean and standard deviation of OC for different BB emission
inventories in the PSEA region was 1.09 ± 0.83 Tg/M, with a coefficient of variation (CV) of 76% (CV is defined as the ratio
of the standard deviation to the mean of all inventories).
Figure 4 illustrates the total emissions of the eight emission inventories in the PSEA region during March 2019 added to the
WRF-Chem after processing (Figure 2). It also presents the percentage composition of CO, OVOCs, NMHCs, NO$_X$, Gas (SO$_2$



and $NH_3$), $PM_{2.5}$, $PM_{10}$, BC, and OC. The total BB emissions (aerosol and gas) are ranked as FINN2.5 MOSVIS (105.7 Tg/M), FINN2.5 MOS (83.7 Tg/M), FINN1.5 (41.9 Tg/M), IS4FIRES (19.4 Tg/M), FEER (15.4 Tg/M), QFED (11.1 Tg/M), GFED (10.3 Tg/M), and GFAS (9.9 Tg/M). Although the total QFED emissions are low, the aerosol emissions (OC, BC, $PM_{2.5}$, $PM_{10}$) are not, just smaller than the FINN schemes. The PSEA aerosol emissions from FINN2.5 are higher than those predicted for FINN1.5 and approximately twice as high as the latter, consistent with the findings of Wiedinmyer et al. (2023). Among them, the highest and lowest emissions of OC+BC are observed in FINN2.5 MOSVIS (2.82 Tg/M) and GFAS (0.32 Tg/M), respectively. Since the FINN schemes employ the EF from Akagi et al. (2011) and subsequent updates, the proportions of each species are relatively similar. In summary, FINN schemes (v1.5 and 2.5) have relatively high total aerosol emissions compared to the other schemes, and the "top-down" scenario (GFAS, FEER, QFED, IS4FIRES) does not have high total emissions despite being constrained by the AOD. To evaluate the spatiotemporal distribution characteristics of absorbing aerosols from BB emissions, particularly the BC to OC ratio, was also displayed in Figure 4. Except for QFED, which exhibits a lower ratio of approximately 0.08 (1/13), the ratios for the other BB datasets are greater than or equal to 0.1(1/10).

**3.2 Model Validation**

To assess the AOPs and DRF simulated by the WRF-Chem adding different BB emissions, the stability of the model is verified by comparing the simulated meteorological fields and $PM_{2.5}$ concentrations with observations at monitoring stations using the WRF-Chem with the FINN1.5 scheme. The statistical results in Table S6 demonstrate good agreement (IOA $\geq$ 0.6) between the simulated T2, RH2, and WS10 and the data from 13 stations. However, at some stations, the wind speed RMSE exceeds 2 m/s, which may be attributed to unresolved topographic features in the surface drag parameterization (Saide et al., 2016). The bias between observations and simulations for RH2 can be partially explained by the influence of different surface and boundary layer parameterizations on the simulated near-surface water vapor fluxes (Chen et al., 2019). During the wildfire period of March 2019, the daily average observed $PM_{2.5}$ concentrations of 23 cities at the surface were compared with the model results for the FINN1.5 case in Figure S2, where the statistical indicators are shown in Table S7. The WRF-Chem was able to simulate $PM_{2.5}$ concentrations in urban sites located in the high BB emission region of northern Laos (Chiang Rai Mueang in northern Thailand and Jinghong in China) with consistency to the observed data (R of 0.64 and 0.75, respectively), where the model was able to reproduce the pollution peaks (IOA of 0.74 and 0.82, respectively). In a previous study by Vongruang et al. (2017), the WRF-CMAQ model was used to simulate $PM_{2.5}$ in the PSEA region by incorporating BB emissions (GFAS v1.1 or FINN1.5) and comparing them with observed stations. The average IOA value was 0.51 (with the optimal IOA being 0.69). In this study, all 23 stations had IOA values greater than 0.51 (with over 52% exceeding 0.69), indicating that the model can consistently reproduce the spatial and temporal distribution characteristics of pollutants in the PSEA region. Although the WRF-Chem model could reasonably capture the spatial-temporal characteristics of $PM_{2.5}$ concentrations observed in most cities (IOA > 0.54), the influence of anthropogenic emission inventories and BB vertical transport may lead to biases in some areas (e.g., Hong Kong).





**3.3 AOD**

**3.3.1 Satellites vs. AERONET AOD**

The linear regression results between AOD daily averages from different satellite sensors and AERONET data are shown in Figure S3. Overall, during the wildfire event in the PSEA region, the DB algorithm of VIIRS demonstrated the best skill, as indicated by optimal $R^2$ and RMSE values. Su et al. (2022) found that VIIRS DB also exhibited the highest accuracy and stability when analyzing long-term multiple satellite inversions of AOD aerosol datasets in Asia. This is because VIIRS DB incorporates upgraded surface and aerosol models specifically designed for Asian regions, which have not been applied to the MODIS DB (Sayer et al., 2019). Therefore, to evaluate the representation of AOD in the WRF-Chem experiments for the PSEA wildfires in March 2019, the AOD at 550 nm provided by VIIRS DB (along with AERONET observations) was chosen to determine biases and errors in the conducted experiments.

**3.3.2 WRF-Chem vs. VIIRS AOD**

To assess the agreement between the simulated AOD from WRF-Chem and the observed AOD, we utilized the extracted data (WRF-Chem) based on VIIRS satellite transit time and compared the daily average values with AERONET observations. Figure 5 illustrates the daily average AOD at 550 nm from the VIIRS and wind (scaled in 10 m/s) at 900 hPa (a), along with the corresponding AOD from the WRF-Chem simulation over the PSEA region during March 2019, considering different BB scenarios (b-i). The high AOD (HAOD, AOD > 1.0) derived from VIIRS retrievals is primarily concentrated in Laos, Thailand, and Vietnam (97-110°E, 15-22.5°N). Additionally, Beibu Gulf and coastal cities in southern China also exhibit high AOD values (AOD > 0.6), which may be attributed to the long-range BB transport of tropical westerly and southwesterly winds depicted in Figure 5(a). The FINN (v1.5 and 2.5), FEER, QFED, and IS4FIRES schemes demonstrate the ability to reproduce high aerosol concentrations in areas with elevated AOD values as observed by VIIRS satellites. These simulations align with the spatial distribution of monthly mean AOD during the wildfire period in the PSEA simulations conducted by Dong and Fu (2015b). However, the GFED and GFAS schemes fail to capture the high AOD areas in the PSEA region, likely due to the low BB emission inventories of the input model (Pan et al., 2020).

Figure 6 ((a)-1 to (a)-8) displays the estimated MB between the model with eight BB scenarios and VIIRS daily mean AOD. The FINN schemes (v1.5 and 2.5) noticeably overestimate AOD in the HAOD region, while the GFED, GFAS, FEER, and IS4FIRES schemes underestimate AOD. Moreover, the FINN schemes also exhibit AOD overestimation in the Beibu Gulf, South China Sea, Bay of Bengal, and Andaman Sea. As the FINN schemes have the largest aerosol emissions compared to other BB emissions (Figure 4), it may lead to an overestimation of AOD in the HAOD region. All schemes exhibit varying degrees of overestimation for a significant portion of southern China. Table 2 provides statistics on the MB of AOD between satellite-retrieved and WRF-Chem AOD in the HAOD region. The AOD simulated by FINN schemes are significantly overestimated, whereas the rest of the schemes exhibit underestimation. Although FEER (-0.12) and IS4FIRE (-0.14) underestimate the simulated AOD, their performance is considerably better than other BB emission inventories. As highlighted





by Palacios-Pena et al. (2017) and Crippa et al. (2019), the MB between simulated and observed AOD can be attributed to
estimation errors in BB uncertainty, aerosol dry mass, and specifically related to the certain mass of small particles or too
much moisture associated with the aerosol. The RMSE estimation (Figure 6(b)-1 to (b)-8) reveals noticeable uncertainty in the
FINN schemes compared to other schemes in the HAOD and southern China, while the performance of the remaining schemes
in simulating AOD in Laos and northern Thailand is unsatisfactory. The RMSE statistics in Table 2 show that the AOD
simulated by the FINN2.5 schemes (MOS and MOSVIS) have greater uncertainty in the HAOD region compared to FINN1.5,
and the RMSE of the other schemes are generally comparable. Figure 6(c)-1 to (c)-8 depicts the temporal R between simulated
AOD and observations, with high values of R (>0.6) concentrated in Laos and northern Thailand, Myanmar, the Bay of Bengal,
the Andaman Sea, and the South China Sea. The FINN2.5 MOSVIS scheme exhibits the highest R compared to other schemes
in the HAOD region (Table 2), potentially due to the updated acquisition time (local time) and increased VIIRS data, leading
to improved R with the observed data.

### 3.3.3 WRF-Chem vs. AERONET AOD

Figure 7 illustrates the time series of AOD at 550 nm, measured at the 16 AERONET sites marked in Figure 1, in comparison
to simulated AOD from WRF-Chem with different BB emissions. These 16 sites are categorized into three major classes,
namely, the satellite inversion of HAOD regions (97-110°E, 15-22.5°N, Figure 7 a-g), the adjacent HAOD area (AHAOD,
Figure 7 h-l), and the downwind area (DA, Figure 7 m-p), allowing for further analysis of AOD variations during wildfire
events. In the HAOD stations (Laos, Chiang Mai, Fang, Nong Khai, Son La, and Ubon Ratchathani), high aerosol loading was
captured by all schemes and AERONET sites on March 15, 23, and 30, respectively. Among the sites, the Laos station
performed the best in terms of simulated and observed AOD mean R and IOA for all BB scenarios, with R and IOA values of
0.82 and 0.80, respectively (Table 3). To compare the performance of the multi-BB emission scenario model for the AOD
simulation, a Taylor diagram was constructed (Figure 8). The Taylor diagram demonstrates that, in the HAOD regions, the
FINN schemes (v1.5 and 2.5) exhibit a higher overall R compared to other schemes when simulating AOD against observations.
Furthermore, the FINN2.5 schemes show a slightly better correlation than FINN1.5. Among the eight schemes, the IS4FIRES
and FINN1.5 schemes simulated AOD performed better in terms of consistency and deviation from the observed comparison
in the HAOD region (Figure 8(a)). In the AHAOD stations, peaks of AOD simulated by WRF-Chem were also found on three
dates (March 15, 23, and 30), but these peaks were lower than the HAOD in Figure 7. Despite the FINN2.5 MOSVIS scheme
showing the best correlation between simulated AOD and observations in the HAOD regions compared to other schemes, its
performance in AHAOD regions was unsatisfactory (Table 3). Poorly performing stations in the AHAOD regions included
Bangkok, Silpakorn, and Songkhla, which are located between 0° and 22.5° N latitude (Figure 7). This discrepancy may be
attributed to the assumptions made by the FINN2.5 MOSVIS scheme for fire detection in the equatorial region to achieve daily
global coverage (Wiedinmyer et al., 2023) and the overestimation of AOD values by WRF-Chem, which can be explained by
the presence of excess aerosol dry mass (Chapman et al., 2009). In the DA regions, such as Hong Kong and Taiwan, high





concentrations of aerosols were simulated and observed after March 23 in Figure 7. Previously, studied the same event using
models and ground measurements and reported a contribution of BB of about 56% to local AOD and 26%-62% to DA.
**3.4 AAOD**
**3.4.1 WRF-Chem vs. TROPOMI AAOD**
Wildfire releases significant amounts of absorbing aerosols such as OC and BC, which can absorb solar radiation and increase
the radiation absorption capacity of the atmosphere, thereby affecting the Earth's radiation balance. Therefore, it is crucial to
evaluate the model's ability to simulate absorbing aerosols using AAOD results obtained from satellite observations. To reduce
the discrepancies caused by missing data in the inversion of different observations, the WRF-Chem simulations are matched
with the observed data. Figure 9 shows the spatial distribution of daily mean AAOD at 500 nm retrieved by TROPOMI (a)
and simulated by WRF-Chem with eight BB emissions (b-j) during March 2019 in the PSEA region. The high AAOD (AAOD >
0.03) from TROPOMI is mainly concentrated in northern Laos, northern Vietnam and northern Thailand, and eastern Vietnam,
which is similar to the spatial distribution characteristics of HAOD provided by VIIRS. Kang et al. (2017) also found similar
AAOD distribution patterns when studying the spatial and temporal characteristics of absorbing aerosols in Southeast Asia
from 2005 to 2016. The WRF-Chem simulations with different BB emissions exhibit high AAOD values not only in the
aforementioned regions but also in southern China and the South China Sea (Figure 9). Figure 10 shows the spatial distribution
characteristics of MB(a), RMSE(b), and R(c) for the comparison of TROPOMI-inverted AAOD with WRF-Chem-simulated
AAOD using different BB scenarios. All FINN, FEER, and IS4FIRES schemes overestimate AAOD in the HAOD region (97-
110°E, 15-22.5°N) compared to TROPOMI inversion, with FINN2.5 showing the most significant overestimation (Figure
10(a)-1 to (a)-8). Table 2 further confirms these overestimations with statistics of 0.056, 0.073, 0.08, 0.02, and 0.018,
respectively. The overestimation may arise from underestimating AAOD in TROPOMI, as well as overestimating absorbing
aerosols in the BB inventory and uncertainties in the representation of absorbing aerosols by WRF-Chem, including aerosol
size distribution, chemical composition, aging processes, vertical and horizontal transport (including injection heights for fire
emissions), and errors in dry/wet removal from the atmosphere. Figure 10(b)-1 to (b)-8 and Table 2 demonstrate that the FINN
schemes exhibit greater uncertainties in simulating AAOD in the HAOD region compared to other schemes. Comparing the R
between satellite-retrieved AAOD and simulated AAOD, values of R > 0.6 are primarily concentrated in northern Laos,
northern Thailand, and Myanmar. Particularly, the FINN2.5 MOSVIS scheme, due to the incorporation of improved local time
and inclusion of small fires from VIIRS, exhibits the best correlation with the simulated AAOD relative to satellite retrievals
(Table 2).
**3.4.2 WRF-Chem vs. AERONET AAOD**
To reduce the uncertainty caused by missing AERONET data, quality control has been applied to the AERONET site data
(samples > 10 days). In the HAOD region within the range of 97-110°E, 15-22.5°N, where both the satellite-retrieved AOD





and AAOD exceed the thresholds of 1 and 0.03 (BB high emission area), respectively. Figure 11 presents a comparison of
time series between AAOD measurements from four AERONET sites within the HAOD region and AAOD simulated by the
nearest corresponding AERONET site using WRF-Chem with different BB inventories. Similar to peaks of AOD, AAOD
from the Doi Ang Khang site also exhibits peaks on March 15th, 23rd, and 30th. Although most schemes can capture the high
AAOD loading, the performances of the GFED, GFAS, and QFED schemes are unsatisfactory (Table S9). This could be
attributed to lower concentrations of absorbing aerosols or inaccurate spatial distribution in the BB emission inventories
(Reddington et al., 2016). The Fang site shows the best mean R and IOA among the eight BB scenarios simulating AAOD
compared with AERONET, with R and IOA values of 0.69 (Table S9). The Taylor diagram indicates that the FINN schemes
perform better than others in representing AAOD in Figure 8 (b), which may be the FINN schemes for unique calculating
biomass burned area and EF that are more suitable for the HAOD region (Wiedinmyer et al., 2011; 2023). When comparing
simulated AAOD with observations for the FINN2.5 MOSVIS scheme, both the R and IOA perform better than other schemes
at all sites. The improved performance of the FINN2.5 MOSVIS scheme in simulating AAOD during wildfires in the PSEA
region can be attributed to two factors: the inclusion of smaller fires using VIIRS 375m fire detection data and updated
information on time and burned area.

## 3.5 AEC

Although AOD and AAOD provide useful information about atmospheric aerosol loading, there is limited information
available regarding the vertical distribution of aerosols. Palacios-Peña et al. (2018) found that uncertainty in the vertical
distribution of aerosols during wildfires in Europe affects AOPs. The CALIPSO, with its unique capability to actively retrieve
vertical aerosol spatial distribution, offers an opportunity to assess the simulation of aerosol vertical optical properties by
WRF-Chem during wildfire events. Figure 12 displays the aerosol vertical extinction profiles at 532 nm retrieved by CALIPSO
in the HAOD region during March 2019, along with the aerosol extinction profiles (550 nm) simulated by various BB schemes,
where model data are matched with CALIPSO overpass times. AEC retrieval by CALIPSO is greater than 0.2 within the range
of 0.5 km to 4 km above ground level, possibly due to the uplifted aerosols from wildfires. WRF-Chem utilizes the smoke
plume rise model, with upper and lower limits of heat flux determined for each land type, to calculate the minimum and
maximum plume heights, and the emitted pollutants are distributed across each vertical layer within the injection height (Grell
et al., 2011). From 0.5 km to 4 km, the trends of AEC changes in the eight BB schemes are consistent with CALIPSO,
indicating that the employed smoke plume rise model in WRF-Chem can reproduce the minimum and maximum plume heights.
However, all the FINN schemes overestimate AEC compared to CALIPSO from 0.5 km to 4 km, while the other schemes
underestimate it. The aerosol concentration in the BB emission inventories may play a decisive role, leading to differences in
the AEC (Reddington et al., 2019). At 4-8 km, the AEC gradually tends to zero with increasing altitude, while the AEC from
the CALIPSO still has three peaks, which may be due to the uncertainty of the model for the BB injection height calculation
or the effect of external dust transmission (Dong and Fu, 2015a; Jin et al., 2022).





**3.6 DRF**

Considering the significant impact of BB aerosols on radiation, this study investigates the radiative perturbation of SW radiation caused by BB aerosols under clear-sky conditions at the top of the atmosphere (TOA), surface (SFC), and in the atmosphere (ATM). The focus is on the DRF of BB aerosols during the daytime, as Ge et al. (2014) found that local convergence in the smoke source region caused by smoke during the daytime transmits more smoke particles on the above surface. Figure 13 illustrates the spatial distribution of daytime average SW radiative perturbation caused by BB aerosols during 2019 March in the PSEA region at the TOA, ATM, and SFC. It is evident that BB aerosol DRF exists not only in the PSEA region but also in other regions such as southern China, Hong Kong, and Taiwan. The spatial distribution of SW radiative perturbation by BB aerosols aligns with the simulated distribution of AOD, with the highest values observed in the HAOD region (97-110°E, 15-22.5°N). Lin et al. (2014) have confirmed that BB aerosols, mainly BC and OC, play significant roles in the radiative budget. On one hand, the solar absorption by BC in the atmosphere increases the rate of radiative heating, leading to a significant decrease in solar radiation reaching the surface. On the other hand, OC enhances the reflected solar radiation at the TOA, resulting in a cooling effect due to reduced incident solar radiation on the atmosphere and surface. The SW radiative perturbation of BB in TOA is negative with a cooling effect in the model domain for eight scenarios, except for areas with high surface albedo such as Himalayan glaciers. Figure 14 shows that during the wildfire period in the HAOD region, the eight schemes exhibit DRF of -30.89±23.6 W/m$^2$ at TOA. The SW radiative perturbation of BB aerosol at TOA depends largely on the SW absorption rate of BB aerosol. The FINN schemes (v1.5 and 2.5) exhibit a significantly stronger cooling effect compared to other schemes, possibly due to higher BC concentrations in BB emissions compared to other inventories. At the ATM, the absorption by BB aerosols leads to a positive radiative forcing, causing atmospheric warming, particularly in the HAOD region. In the HAOD region, the eight schemes exhibit a BB aerosol SW DRF of 1.70±1.40 W/m$^2$ in the ATM (Figure 14). WRF-Chem can simulate the heating effect of BB aerosols in the ATM regardless of the BC/OC ratio used in the emission inventory (1:8, 1:9, or 1:13). At the SFC, the cooling effect is due to the scattering of non-absorbing atmospheric aerosols and absorbing aerosols that increase the radiative heating rate, resulting in a significant reduction of solar radiation reaching the surface. The eight schemes simulate the DRF of -32.60±24.50 W/m$^2$ at SFC in the daytime (with FINN2.5 MOSVIS reaching a maximum of 70 W/m$^2$), which is comparable to the level of the PSEA region studied previously by Lin et al. (2014) and Ge et al. (2014).

**4. Discussion**

Biases in the simulated AOPs (AOD, AAOD, AEC) over tropical BB have been attributed to a variety of factors (Reddington et al., 2016), including (1) uncertainties in BB emission fluxes, (2) errors in modeling the atmospheric distribution and properties of BB aerosols. These deviations in optical properties further affect the DRF, leading to uncertainties in the assessment of climate change.





### 4.1 BB Emission Fluxes

Uncertainties associated with the derivation of emission fluxes arise from errors in satellite detection of active fire or burned areas (e.g., cloud and smoke obscuration of the surface, satellite spatial resolution and detection limitations, and satellite exceedance times), as well as uncertainties in EF and fuel consumption estimates (Carter et al., 2020; Wiedinmyer et al., 2023). Eight BB inventories were inverted from MODIS data, but there were significant gaps between the bandwidths of MODIS in the equatorial region, as well as difficulties in detecting fires located under thick clouds, and a reduction in fire detection sensitivity at the scan edge sensitivity, leading to an underestimation of total regional BB emissions (Wang et al., 2018). In this paper, The FINN2.5 dataset (BB emission fluxes and AOPs) is consistently higher than the other datasets, with FINN2.5 MOSVIS being the highest overall. FINN2.5 includes improved burned area calculations, uses year-specific land cover and vegetation datasets, updates fuel loads and EF, and can use multiple fire detection satellite inputs (e.g., MODIS and VIIRS), which may account for the improved BB emission fluxes. In the PSEA region, during wildfire events, the BB emissions from FINNv2.5 are consistently higher than the emissions provided by FINNv1.5, approximately twice as much as the latter, even when considering only MODIS fire detections. The increase in emissions is primarily attributed to the new treatment of burned areas (Wiedinmyer et al., 2023). Despite updates to input data, parameters, and processing methods, the FINN2.5 scheme tends to overestimate AOPs compared to observations. This overestimation may arise from inaccurate ecosystem identification (e.g., tropical forests instead of shrublands or areas with fewer trees) and fuel load allocation (Pan et al., 2020). Furthermore, in tropical regions, the FINN scheme employs smoothing of fire detections to mitigate the impact of clouds, which could lead to an overestimate of BB emissions (Wiedinmyer et al., 2011; 2023). QFED provides relatively higher OC concentrations, but lower total BB emissions, and the primary driving factors behind these differences are the assumed fuel types and related EF. Therefore, it is inappropriate to consider OC as the sole criterion for evaluating BB emission fluxes when comparing multiple BB emission inventories. Although the aerosol concentrations provided by QFED are larger than those of IS4FIRES and FEER, the simulated AOPs and DRF of this scheme are lower than those of the latter, which may be due to the influence of secondary pollutant emission precursors ($NO_2$, $NH_3$, etc.). Previous studies have often used an expansion of aerosols (BC+OC) in the BB emission inventories by a factor of 3-6 to assess the AOPs (Reddington et al., 2016; Marlier et al., 2013), and the simulation results from the QFED scheme above reveal that there may be significant uncertainties in this expanded aerosol (BC+OC) approach. Although GFED4.1s improves the detection of small fires, the agricultural EF = 2.3 g/kg is lower than in other emission inventories, which could result in an underestimation of AOPs simulated by WRF-Chem with the GFED scheme. Yin (2020) found that BB in the PSEA region from 2001 to 2018 was predominantly driven by agro-residue burning and shrubland fires while GFED4.1s underestimation of DM for both fires and the mismatch in vegetation types may have contributed to the underestimation of BB emission fluxes (Reddington et al., 2016). In general, FRP-based estimation methods, such as GFAS, FEER, QFED, and IS4FIRES, allow for a more direct estimation of fuel consumption from fire-release energy without the uncertainty associated with the estimation. However, in the PSEA region, when the FRP from MODIS inversion



is observed at a nominal spatial resolution of 1 km at its nadir, it risks missing a large number of smaller fires, as well as
missing fires that are obscured by clouds (Dong and Fu, 2015b), which may lead to an underestimation of the simulated AOPs.

**4.2 Modeling Uncertainty and Calculation Bias**

The representation of BB aerosols in the model is inadequate, including insufficient characterization of aerosol size distribution,
chemical composition, aging processes, vertical and horizontal transport (including injection heights of fire emissions), and
model errors in dry/wet deposition from the atmosphere (Palacios-Peña et al., 2018; Reddington et al., 2016). Sensitivity
experiments using the global aerosol model reveal that calculations of hygroscopicity growth are most sensitive in simulating
AOD (Reddington et al., 2016). The contribution of SOA formed through the oxidation of VOCs in BB plumes is also a
significant source of uncertainty (Jathar et al., 2014). In this study, we employed the meteorological chemistry and aerosol
scheme: MOZART-MOSAIC_4bin_aqueous, which includes aqueous-phase chemistry and SOA, but this mechanism may
lead to overestimation/underestimation of AOPs in the model. The smoke plume rise model developed by Freitas et al. (2010)
was used to vertically represent smoke plumes. Although all schemes capture the vertical profiles of BB aerosol extinction
from 0.5 km to 4 km altitude, some deviations still exist. Previous research has indicated that assuming all fire emissions
injected at the top of the plume could be a worse assumption than prescribing surface-based emissions, which may lead to
deviations in simulated AOPs (Mallia et al., 2018). The AEC is not characterized in all BB scenario simulations for 4-8 km,
which may also lead to an underestimation of AOD or AAOD, and this high-level perturbation of AEC may come from the
influence of external dust aerosols, so the model emission inventory should consider the effect of dust emissions. Others studies
have also found that uncertainties in anthropogenic emission inventories can also lead to simulation errors in AOPs and DRF
during wildfires in the PSEA region (Dong and Fu, 2015a). Although we used the latest version of EDGAR 2015 data, there
may be some underestimation of such emission inventories with a large number of incoming factories in the PSEA region
(Yang, 2016). Additionally, the inclusion of direct and indirect radiation feedback in the WRF-Chem model has been found to
effectively improve the simulation of AOPs in European wildfire simulations (Palacios-Peña et al., 2019), whereas this study
only incorporates aerosol direct radiative feedback. In the calculation of the AOPs, uncertainties associated with the optical
properties of the assumed BB aerosols, such as their refractive index, may also lead to biases in the AOPs. There is some
uncertainty in the AOD from the VIIRS satellite inversion and in the SSA and AAOD from the TROPOMI inversion due to
cloud cover effects in the PSEA region, which may also lead to biased assessments. In addition, the closest proximity method
used in the gridding process of BB emission inventories can also lead to some calculation errors.

**5. Summary and Conclusion**

This study conducted sensitivity analyses to simulate AOPs and DRF in the PSEA region using eight commonly global BB
emission inventories (GFED, FINN1.5, FINN2.5 MOS, FINN2.5 MOSVIS, GFAS, FEER, QFED, IS4FIRES) and the WRF-
Chem model. The main findings can be summarized below.





Regarding BB emissions in the PSEA region, high OC emissions in all datasets (BB) are mainly concentrated in the northern
parts of Laos, Cambodia, and Thailand, and in eastern Myanmar, with a difference in emissions of about a factor of 9 (0.295-
2.533 Tg/M), an overall mean and standard deviation of 1.09±0.83 Tg/M and a CV of 76%, respectively. Those high BB
emissions are primarily from savanna and agricultural fires. OC emissions in GFED and GFAS are significantly lower than in
the other inventories. This is attributed to lower DM and agricultural fire EF in GFED, while DM is underestimated in GFAS.
The OC in FINN2.5 VISMOS is about twice as high as those in FINN1.5, which is explained by the difference in DM rather
than EF. Total aerosol emissions are relatively high in the FINN scenarios (v1.5 and 2.5) compared to the other scenarios.
Although the "top-down" emission inventories (GFAS, FEER, QFED, IS4FIRES) are constrained by the AOD from MODIS,
the total aerosol emission flux is still insufficient.
The AOD from VIIRS (DB algorithm) demonstrates the best ability to retrieve the AOD compared to AERONET data. An
evaluation of the AOPs in the PSEA region during March 2019 reveals different performances between observations (VIIRS,
TROPOMI, AERONET) and BB emission inventories. When comparing the AOD simulated by WRF-Chem with the observed
AOD from VIIRS, the FINN1.5, FEER, QFED, and IS4FIRES schemes show a better ability to reproduce high aerosol
concentrations in the HAOD region, the GFED and GFAS schemes show limitations in characterizing these regions. The FINN
(v1.5 and 2.5) schemes tend to overestimate AOD in the region, while other schemes underestimate AOD. The comparison
with AERONET data further highlights the performance of different BB emission scenarios, with the FINN1.5 and IS4FIRES
scenarios generally showing better agreement with observations. For AAOD comparison, it was found that the WRF-Chem
simulations with different BB scenarios were less capable of simulating AAOD than AOD. The unsatisfactory performance of
the GFED, GFAS, and QFED schemes may be due to low concentrations of absorbing aerosols or inaccuracies in the spatial
distribution of BB emissions. Among the evaluated BB scenarios, the FINN1.5 schemes generally performed better in
representing AAOD. Particularly, the FINN2.5 MOSVIS scheme, due to the incorporation of improved local time and inclusion
of small fires from VIIRS, exhibits the best R with the simulated AOD and AAOD relative to observations. CALIPSO
observations versus AEC simulated by WRF-Chem suggest that the smoke plume rise model can reproduce the minimum and
maximum smoke plume heights of wildfire aerosols. However, the FINN (v1.5 and 2.5) schemes tend to overestimate the AEC
compared to CALIPSO, while the other scenarios underestimate it. Regarding the DRF, the spatial distribution of the SW
radiative disturbances due to BB aerosols closely follows the pattern of the AOD. the FINN (v1.5 and 2.5) schemes exhibit a
stronger cooling effect at TOA, which may be due to the higher BC concentration in its emissions. In the HAOD region, BB
aerosols exhibited a daytime SW radiative forcing of -32.60±24.50 W/m$^2$ at the SFC, positive forcing (1.70±1.40 W/m$^2$) in the
ATM, and negative forcing (-30.89±23.6 W/m$^2$) at the TOA. Overall, the FINN scenarios (especially FINN2.5) result in an
overestimation of the AOPs in the PSEA region due to an overestimation of DM rather than EF, which in turn may lead to an
overestimation of the DRF. Although the FINN2.5 MOSVIS scenario presents an overestimation of AOPs, the R is the best.
Although the "top-down" emission inventory (GFAS, FEER, QFED, IS4FIRES) is constrained by the AOD from MODIS, the
total aerosol emission flux is still insufficient, which leads to an underestimation of the AOPs modeled by WRF-Chem in the
PSEA region. In addition, uncertainties in anthropogenic emissions, dust emissions, and vertical distribution of aerosol





concentrations, may be attributed to differences from simulations versus observations during the wildfire period in the PSEA region.

Additional evaluations of satellite-based fire emission inventories, particularly in large BB source regions (PSEA), would contribute to a deeper understanding of the uncertainties associated with fire emissions. In the PSEA region, greater attention should be given to the impacts of small fires, cloud cover, different ecosystem types, and EF during various burning stages and ecosystem types on the inversion of BB emission inventories. To further explore the subsequent effects of BB emissions (e.g., AOPs and radiative forcing), additional investigation of fire aerosol aging and treatment uncertainties (e.g., injection height, mixing state, SOA formation) are needed. Our study demonstrates that the uncertainty in BB emission inventories is an important factor influencing the WRF-Chem simulation of air quality and climate during wildfires, although the limitations of the model itself should not be overlooked. In the future, we will conduct additional sensitivity experiments and utilize more observational data to further validate the aforementioned uncertainties.

**Data availability**

Global Fire Emissions Database, Version 4.1 (GFEDv4.1) are available at https://doi.org/10.3334/ORNLDAAC/1293 (Randerson et al., 2017); The Fire INventory from NCAR (FINN, including version 1.5 and 2.5) data files can be downloaded from https://www.acom.ucar.edu/Data/fire/ (Wiedinmyer et al., 2011); CAMS global biomass burning emissions based on fire radiative power (GFAS v1.2) at https://ads.atmosphere.copernicus.eu/cdsapp#!/dataset/cams-global-fire-emissions-gfas (Rémy et al., 2017); Fire Energetics and Emissions Research version 1.0 (FEER) data files can be downloaded from https://feer.gsfc.nasa.gov/data/emissions/ (Ichoku and Ellison, 2014); Quick Fire Emissions Dataset version 2.5 release 1 (QFED) data can be accessed from https://portal.nccs.nasa.gov/datashare/iesa/aerosol/emissions/QFED/v2.5r1/ (Koster et al., 2015), and Integrated Monitoring and Modelling System for Wildland FIRES Project version 2.0 (IS4FIRES) data files can be downloaded from http://silam.fmi.fi/thredds/catalog/i4f20emis-arch/catalog.html (Soares et al., 2015).

**Author contributions**

Conceptualization, methodology, and writing–original draft, Y.B.J.; Y.B.J. and Y.M.L. designed the research framework and collected the materials; Y.B.J. calculated the emissions and drew the figures; Y.M.L. and Y.B.J. analyzed the results and wrote the paper with inputs from all authors; All authors contributed to the discussion and improvement of the paper; Supervision, Q.F.



**Financial support**
This work was supported by the Guangdong Major Project of Basic and Applied Basic Research (Grant No.
2020B0301030004), the National Key Research and Development Program of China (Grant No. 2019YFC0214605), Science
and Technology Program of Guangdong Province (Science and Technology Innovation Platform Category) (Grant No.
2019B121201002), and the National Natural Science Foundation of China (Grant No. 42075181).
**Competing interests**
The authors declare that they have no conflict of interest.

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

**Appendix A**
Abbreviations and Acronyms

| | |
|---|---|
| AAOD | Absorbing aerosol optical depth |
| AEC | Aerosol extinction coefficient |
| AHAOD | Adjacent HAOD area |
| AOD | Aerosol optical depth |
| AOPs | Aerosol optical properties |
| ATM | In the atmosphere |
| BB | Biomass burning |
| BC | Black carbon |
| CALIPSO | Cloud-Aerosol Lidar and Infrared Pathfinder Satellite Observation |
| CAM-chem | Community Atmosphere Model with Chemistry |
| DA | Downwind area |
| DRF | Direct radiative forcing |
| DM | Dry matter |
| EDGAR | Emissions Database for Global Atmospheric Research |
| EF | Emission factors |
| FEER | Fire Energetics and Emissions Research |
| FINN | Fire INventory from NCAR |
| FRP | Fire radiative power |
| GEOS-Chem | Goddard Earth Observing System-Chemistry |
| GFAS | Global Fire Assimilation System |
| GFED | Global Fire Emissions Database |
| HAOD | High AOD |
| IS4FIRES | Integrated Monitoring and Modelling System for Wildland FIRES Project |
| LW | Longwave |
| MEGAN | Model of Emissions of Gases and Aerosols from Nature |
| MODIS | Moderate Resolution Imaging Spectroradiometer |
| MOSAIC | Model for Simulating Aerosol Interactions and Chemistry |



| MOZART | The Model for Ozone and Related chemical Tracers |
| NMHCs | Non-methane hydrocarbons |
| NMVOCs | Non-methane volatile organic compounds |
| OC | Organic carbon |
| OVOCs | Oxygenated volatile organic compounds |
| PSEA | Peninsular Southeast Asia |
| PM | Particulate matter |
| QFED | Quick Fire Emissions Dataset |
| RH2 | 2 m relative humidity |
| SFC | At the surface |
| SOA | Secondary organic aerosol |
| SSA | Single scattering albedo |
| SW | Shortwave |
| T2 | 2 m temperature |
| TOA | The top of the atmosphere |
| TPM | Total particle matter |
| VIIRS | Visible Infrared Imaging Radiometer Suite |
| WS10 | 10 m wind speed |






Tables
**Table 1. Comprehensive comparison of eight BB emission inventories globally in terms of different methodological details and**
**species, where Bottom-up approach to construct emission inventories are GFED v4.1s, FINN v1.5, FINN v2.5 MOS, FINN v2.5**
**MOSVIS, and others are Top-down approach.**

| BB dataset | Resolution Temporal | Data source | Main EF[a] | OVOCs[b] | NMHCs[c] | Gas | Aerosols |
|---|---|---|---|---|---|---|---|
| GFED v4.1s | 0.25°x 0.25° 3-hourly daily monthly 1997-2022 | MODIS C5 | Akagi et al. (2011), Andreae and Merlet (2001) with updates | $CH_3COCHO$, $CH_3COOH$,etc | $C_2H_4$,$C_2H_6$, $C_3H_8$, etc | CO, $NO_x$, $SO_2$, $NH_3$ | OC, BC, $PM_{2.5}$ |
| FINN v1.5 | 1 km² Daily 2002-Present | MODIS C6 | Akagi et al. (2011), Andreae and Merlet (2001) | $CH_3COCHO$, $CH_3COOH$,etc | $C_2H_4$,$C_2H_6$, $C_3H_8$, etc | CO, $NO_x$, $SO_2$, $NH_3$ | OC, BC, $PM_{2.5}$,$PM_{10}$ |
| FINN v2.5 MOS | 1 km² Daily 2002-2021 | MODIS C6 | Akagi et al. (2011), Wiedinmyer et al (2011) | $CH_3COCHO$, $CH_3COOH$,etc | $C_2H_4$,$C_2H_6$, $C_3H_8$, etc | CO, $NO_x$, $SO_2$, $NH_3$ | OC, BC, $PM_{2.5}$, $PM_{10}$ |
| FINN v2.5 MOSVIS | 1 km² Daily 2002-2021 | MODIS C6 VIIRS | Akagi et al. (2011), Wiedinmyer et al (2011) | $CH_3COCHO$, $CH_3COOH$,etc | $C_2H_4$,$C_2H_6$, $C_3H_8$, etc | CO, $NO_x$, $SO_2$, $NH_3$ | OC, BC, $PM_{2.5}$, $PM_{10}$ |
| GFAS v1.2 | 0.1°x 0.1° Daily 2003-Present | MODIS C6 | Akagi et al. (2011) | $CH_3COCHO$, $CH_3COOH$,etc | $C_2H_4$,$C_2H_6$, $C_3H_8$, etc | CO, $NO_x$, $SO_2$, $NH_3$ | OC, BC, $PM_{2.5}$ |
| FEER v1.0-G1.2 | 0.1°x 0.1° Daily 2003-Present | GFAS v1.2 FRP | Andreae and Merlet (2001) | $CH_3COCHO$, $CH_3COOH$,etc | $C_2H_2$,$C_2H_6$, $C_3H_8$, etc | CO, $NO_x$, $SO_2$, $NH_3$ | OC, BC, $PM_{2.5}$ |
| QFED v2.5r1 | 0.1°x 0.1° Daily 2000-Present | MODIS C6 | Akagi et al., (2011), Andreae and Merlet, (2001) | $CH_3COCHO$, $CH_3COOH$,etc | $C_2H_6$,$C_3H_6$, $C_3H_8$, etc | CO, $NO_x$, $SO_2$, $NH_3$ | OC, BC, $PM_{2.5}$ |
| IS4FIRES v2.0 | 0.1°x 0.1° 3-hourly 2000-Present | MODIS C6 | Akagi et al. (2011), Sofiev et al., (2009) | NA | NA | NA | TPM[d] |

**[a]The main references for Emission factors (EF) used in the BB emission database.**
**[b]Oxygenated volatile organic compounds (OVOCs) contain C, H, and O. examples include alcohols, aldehydes, ketones, and organic**
**acids.**
**[c]Non-methane hydrocarbons (NMHCs) are defined as organic compounds excluding methane ($CH_4$) that contain only C and H.**
**[d]The total particle matter (TPM) considers three different particle sizes (0.17 µm, 1.1 µm and 3 µm).**
**Notes: OVOCs and NMHCs together account for nearly all the gas-phase non-methane volatile organic compounds (NMVOC)**
**emitted by fires (Akagi et al., 2011). NA: Not available.**





**Table 2. WRF-Chem AOD and AAOD vs. satellites evaluation in HAOD (97-110°E, 15-22.5°N) region during March 2019.**

| BB Inventories | WRF-Chem vs. VIIRS | | | WRF-Chem vs. TROPOMI | | |
|---|---|---|---|---|---|---|
| | MB | RMSE | R | MB | RMSE | R |
| GFED | -0.26 | 0.48 | 0.22 | 0.009 | 0.018 | 0.191 |
| FINN1.5 | 0.39 | 0.71 | 0.27 | 0.056 | 0.071 | 0.190 |
| FINN2.5 MOS | 0.63 | 0.98 | 0.27 | 0.073 | 0.094 | 0.205 |
| FINN2.5 MOSVIS | 0.78 | 1.01 | 0.28 | 0.080 | 0.102 | 0.232 |
| GFAS | -0.34 | 0.52 | 0.21 | 0.004 | 0.013 | 0.185 |
| FEER | -0.12 | 0.44 | 0.25 | 0.020 | 0.029 | 0.213 |
| QFED | -0.24 | 0.46 | 0.23 | 0.011 | 0.020 | 0.187 |
| IS4FIRES | -0.14 | 0.43 | 0.27 | 0.018 | 0.028 | 0.208 |





**Table 3. WRF-Chem AOD at 550 nm vs. AERONET in HAOD, AHAOD, and DA during the wildfire period, where HAOD includes Laos, Chiang Mai, Doi Ang Khang, Fang, Nong Khai, Son La, and Ubon Ratchathani stations.**

| Stations | Variables | BB emission inventories | | | | | | | |
|---|---|---|---|---|---|---|---|---|---|
| | | GFED | FINN1.5 | FINN2.5 MOS | FINN2.5 MOSVIS | GFAS | FEER | QFED | IS4FIRES |
| Laos | R | 0.74 | 0.9 | 0.9 | 0.81 | 0.7 | 0.84 | 0.79 | 0.85 |
| | IOA | 0.78 | 0.83 | 0.75 | 0.75 | 0.76 | 0.84 | 0.8 | 0.86 |
| Chiang Mai | R | 0.46 | 0.61 | 0.53 | 0.77 | 0.48 | 0.54 | 0.45 | 0.55 |
| | IOA | 0.75 | 0.79 | 0.74 | 0.82 | 0.73 | 0.77 | 0.76 | 0.78 |
| Doi Ang Khang | R | 0.48 | 0.66 | 0.66 | 0.8 | 0.49 | 0.64 | 0.52 | 0.63 |
| | IOA | 0.78 | 0.75 | 0.68 | 0.69 | 0.77 | 0.81 | 0.79 | 0.81 |
| Fang | R | 0.42 | 0.71 | 0.7 | 0.85 | 0.42 | 0.68 | 0.5 | 0.63 |
| | IOA | 0.71 | 0.81 | 0.77 | 0.82 | 0.7 | 0.73 | 0.71 | 0.75 |
| Nong Khai | R | 0.25 | 0.39 | 0.59 | 0.51 | 0.28 | 0.27 | 0.31 | 0.37 |
| | IOA | 0.73 | 0.71 | 0.69 | 0.65 | 0.71 | 0.72 | 0.73 | 0.74 |
| Son La | R | 0.5 | 0.75 | 0.76 | 0.64 | 0.43 | 0.81 | 0.64 | 0.64 |
| | IOA | 0.72 | 0.72 | 0.65 | 0.65 | 0.71 | 0.84 | 0.75 | 0.79 |
| Ubon Ratchathani | R | 0.23 | 0.6 | 0.54 | 0.3 | 0.41 | 0.35 | 0.36 | 0.37 |
| | IOA | 0.68 | 0.64 | 0.61 | 0.58 | 0.64 | 0.69 | 0.66 | 0.69 |
| AHBA | $\bar{R}$ | 0.44 | 0.51 | 0.48 | 0.24 | 0.53 | 0.52 | 0.55 | 0.52 |
| | $\overline{IOA}$ | 0.73 | 0.69 | 0.66 | 0.63 | 0.72 | 0.76 | 0.75 | 0.74 |
| DA | $\bar{R}$ | 0.43 | 0.41 | 0.39 | 0.48 | 0.44 | 0.44 | 0.46 | 0.39 |
| | $\overline{IOA}$ | 0.69 | 0.71 | 0.69 | 0.71 | 0.69 | 0.71 | 0.70 | 0.70 |

Note: AHAOD and DA only contain the corresponding site mean R and IOA



**Figures**



**Figure 1. WRF-Chem simulation domain (D01, blue line), which is mainly the PSEA region (purple line, including Vietnam, Thailand, Myanmar, Cambodia, and Laos), South China Sea, and South China region, where the red dots are AERONET stations, the black crosses are air quality stations, and the blue crosses are meteorological stations.**




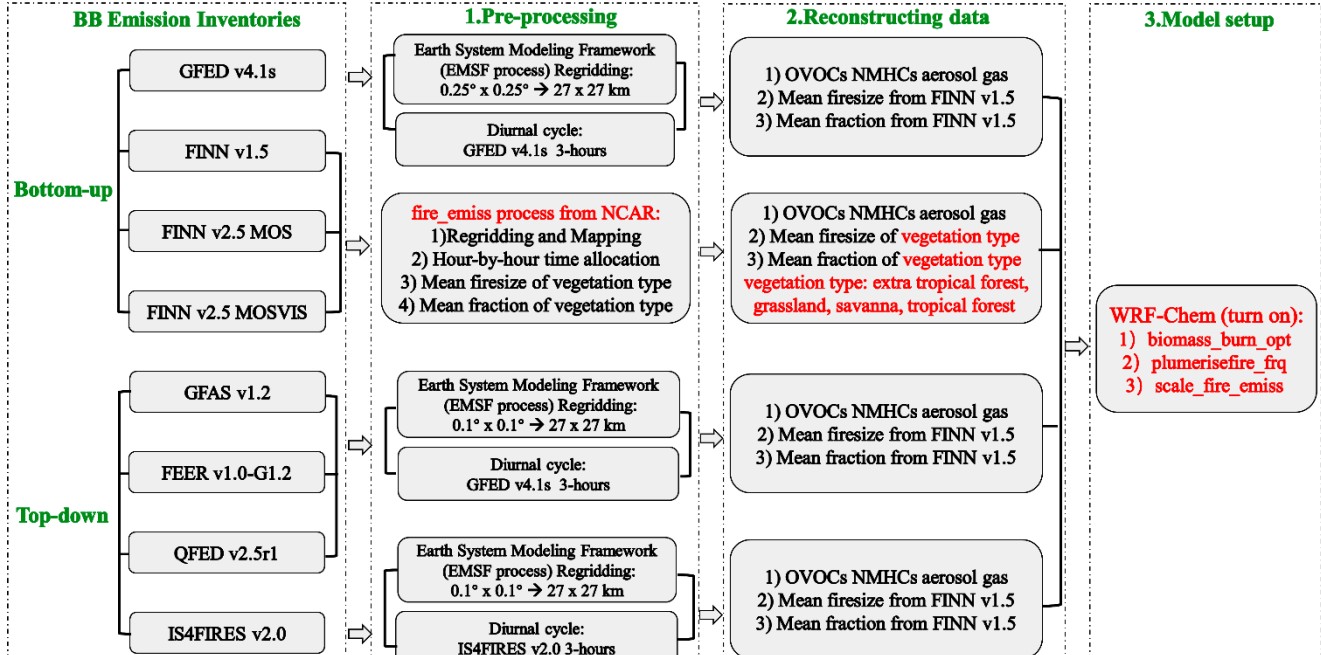

**Figure 2.** The flowchart illustrates the three processes of Pre-processing, Reconstructing data, and Model setup to put the eight BB
emission inventories into the WRF-Chem simulation of AOPs and DRFs during the March 2019 wildfires in the PSEA region. The
Pre-processing consisted of re-gridding and time allocation, where the FINNs scenario was processed using the fire_emiss program
from NCAR, while the grids generated by the other scenarios based on the FINN 1.5 scenario were spatially allocated using the
EMSF program. The GFED, GFAS, FEER, and QFED have the same time allocations as GFED, and the remainder use self-
contained time allocations. The Reconstructing data has three components: emissions (OVOCs, NMHCs, aerosol, and gas) composed
by the MOZART-MOSAIC mechanism, fire size, and vegetation proportions (extratropical forest, grassland, savanna, tropical
forest). Compared to the FINNs scheme, the missing compounds and aerosols from the other schemes were added based on the
methodology of Jose et al. (2017), Andreae and Merlet (2001;2019). Eight BB emission inventories used the fire sizes provided by
the FINN 1.5 scheme, as well as the vegetation proportions. The Model setup turned on BB simulations including the smoke plume
rise.



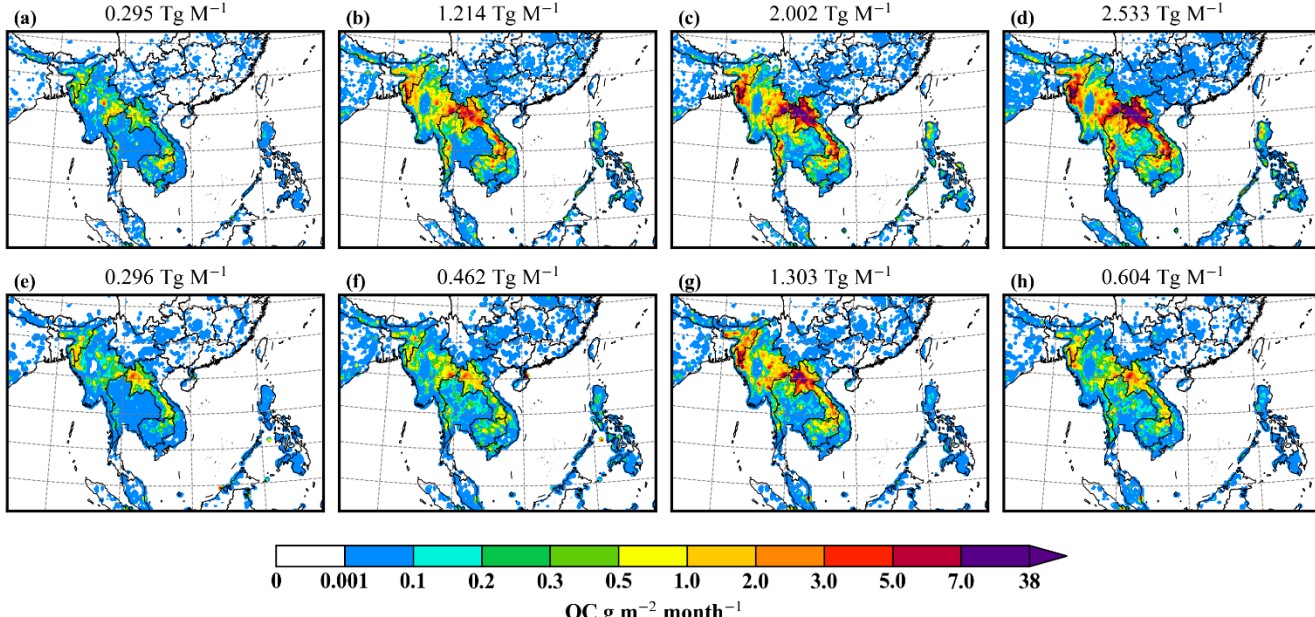


**Figure 3. The spatial distribution of eight BB emission inventories of OC in the study region, for (a-h): GFED, FINN1.5, FINN2.5 MOS, FINN2.5 MOSVIS, GFAS, FEER, QFED, IS4FIRES, and the total OC emissions in the PSEA region during March 2019.**





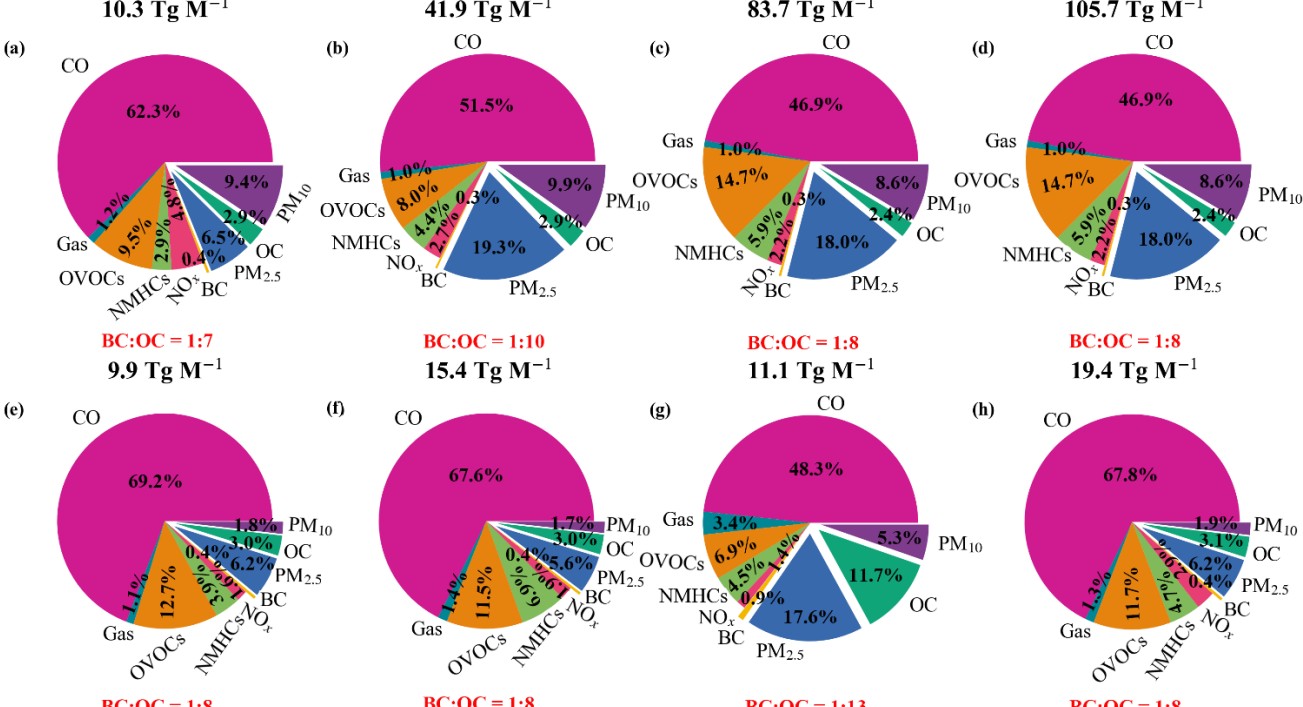

**Figure 4. Total emissions and percentage composition of different substances in the eight BB emission inventories (after processing in Figure 2, i.e., the missing BB data has been supplemented.) over PSEA in the WRF-Chem model, which indicates the proportion of BC and OC, where "Gas" represents the combination of $SO_2$ and $NH_3$. OVOCs contain C, H, and O compounds (ethanol ($C_2H_5OH$), formaldehyde ($CH_2O$), acetaldehyde ($CH_3CHO$), acetone ($CH_3COCH_3$), methanol ($CH3OH$), methyl ethyl ketone (MEK), pentanedial ($C_5H_6O_2$), acetic acid ($CH_3COOH$), cresol ($C_6H_4(CH_3)(OH)$), glyceraldehyde (GLYALD), methanal (Mgly), glyoxal ($CH_3COCHO$), acetol ($CH_3COCH_2OH$), methyl vinyl ketone (MACR), methyl vinyl ketone (MVK)). NMHCs refer to organic compounds containing only C and H besides methane ($CH_4$), including pentane ($C_5H_{12}$), butadiene ($C_4H_8$), ethylene ($C_2H_4$), ethane ($C_2H_6$), propane ($C_3H_8$), propylene ($C_3H_6$), toluene ($C_6H_5(CH_3)$), decane ($C_{10}H_{16}$), isoprene ($C_5H_8$). NMHCs and OVOCs combined constitute nearly all of the non-methane volatile organic compounds (NMVOCs) emitted by wildfires. $PM_{2.5}$ is the $PM_{2.5}$ fraction excluding OC and BC. $PM_{10}$ is the $PM_{10-2.5}$ fraction.**





**Figure 5. The daily mean AOD retrieved by the VIIRS satellite (a) transiting the PSEA region and the AOD simulated by WRF-Chem with eight corresponding BB emission inventories (b-i, GFED, FINN1.5, FINN2.5 MOS, FINN2.5 MOSVIS, GFAS, FEER, QFED, IS4FIRES) in the PSEA region during March 2019, where 950 hPa wind (vectors, m/s) based on March 2019 of ERA5 data.**



**Figure 6. Spatial distribution of MB, RMSE, and R between AOD from VIIRS satellite vs. AOD simulated by WRF-Chem with 8 BB emission inventories (GFED, FINN1.5, FINN2.5 MOS, FINN2.5 MOSVIS, GFAS, FEER, QFED, IS4FIRES) in PSEA during March 2019, where (a)-1 to (a)-8 are the MB for the comparison of the eight BB scenarios, (b)-1 to (b)-8 are the RMSE for the comparison of the eight BB scenarios, (c)-1 to (c)-8 are the R for the comparison of the eight BB scenarios.**



959

**Figure 7. Time series of daily average AOD (550 nm) simulated by WRF-Chem including 8 BB emissions in March 2019 compared to 16 AERONET sites (a-p). These stations are divided into three categories, where the first category of stations is located within the HAOD range of satellite inversion (97-110°E, 15-22.5°N, a-g); The second type consists of observational sites located in adjacent high AOD regions (namely AHAOD, h-l); The third type encompasses observational sites situated within the downwind areas (namely DA, m-p). The legend line characterizes different BB simulation scenarios.**




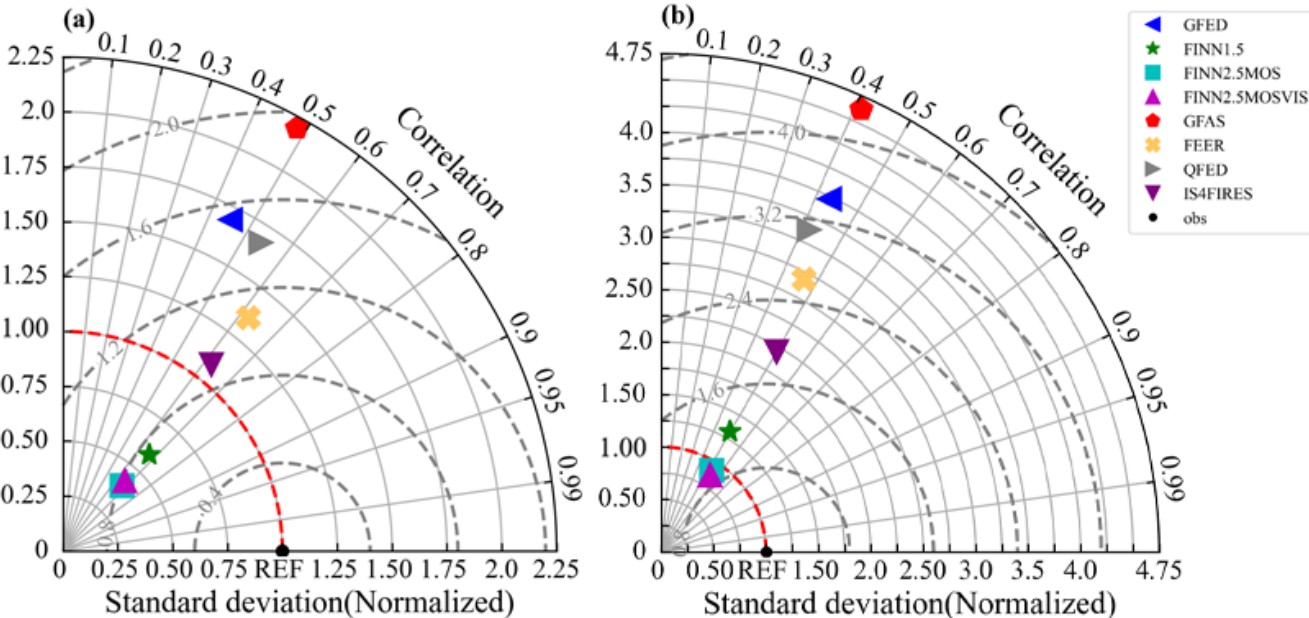


**Figure 8. Taylor diagrams of (a) AERONET vs. WRF-Chem AOD at 550 nm and (b) AERONET vs. WRF-Chem AAOD at 500 nm in the HAOD region (97-110°E, 15-22.5°N) during the wildfire period.**






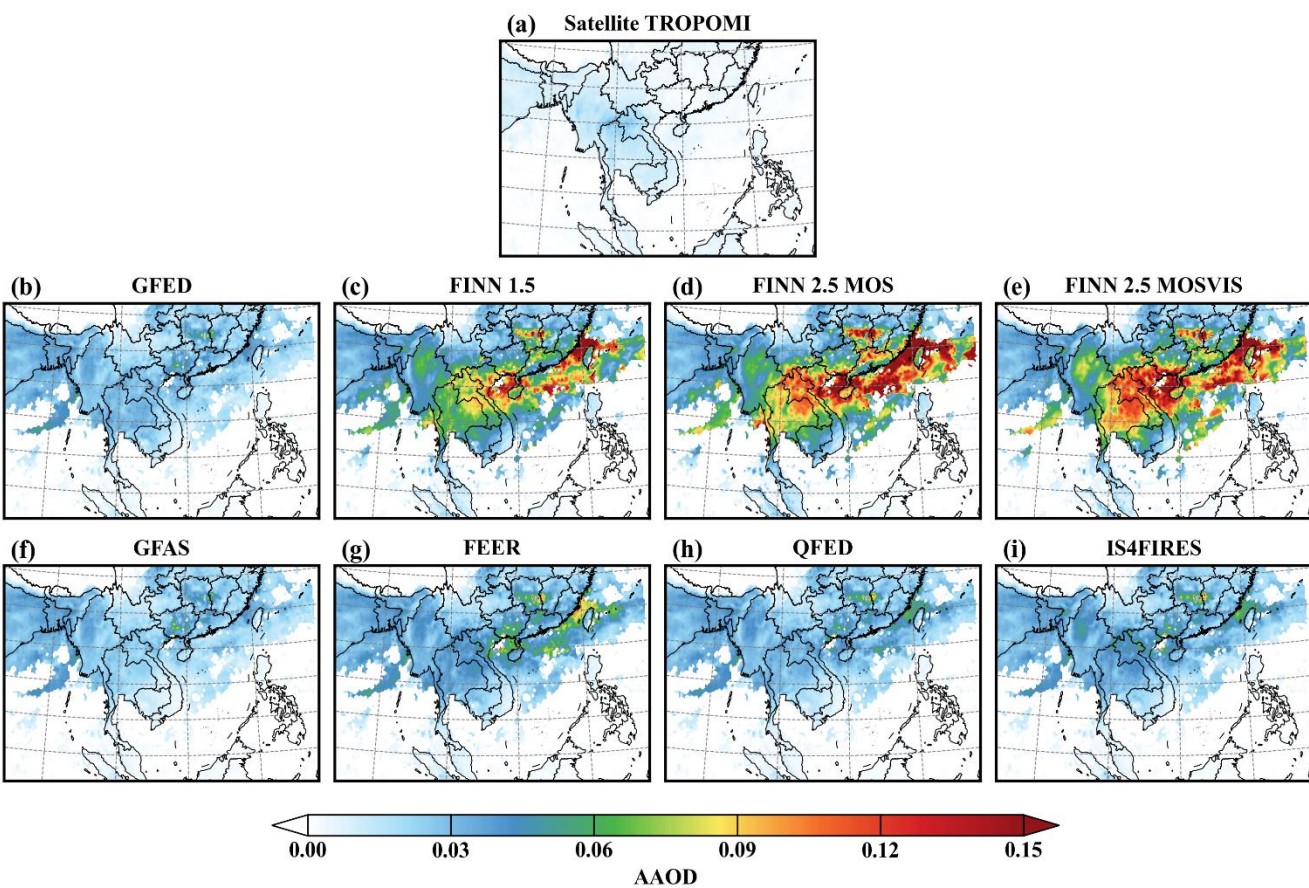


**Figure 9. Spatial distribution of AAOD between Sentinel-5 TROPOMI satellite (a) vs. AAOD simulated by WRF-Chem with 8 BB emission inventories (b-i) during wildfire period in PSEA.**
















**Figure 10. Spatial distribution of MB, RMSE, and R between AOD from VIIRS satellite vs. AOD simulated by WRF-Chem with 8 BB emission inventories (GFED, FINN1.5, FINN2.5 MOS, FINN2.5 MOSVIS, GFAS, FEER, QFED, IS4FIRES) in PSEA during March 2019, where (a)-1 to (a)-8 are the MB for the comparison of the eight BB scenarios, (b)-1 to (b)-8 are the RMSE for the comparison of the eight BB scenarios, (c)-1 to (c)-8 are the R for the comparison of the eight BB scenarios.**





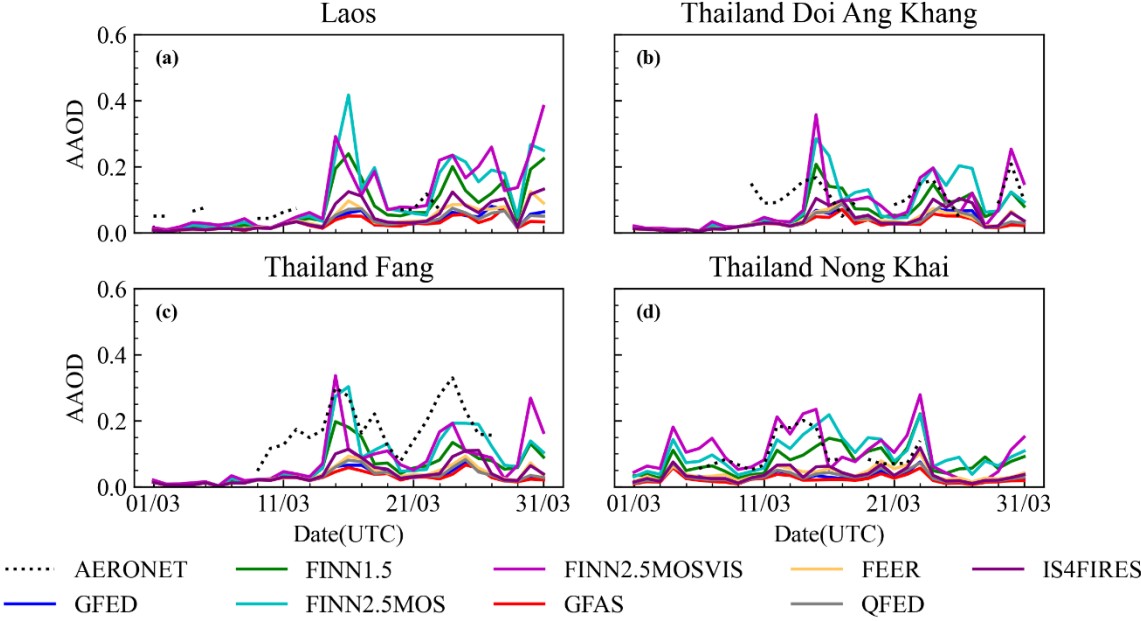

998

**Figure 11. Comparisons of time series between daily mean AAOD at 500 nm measurements provided by four AERONET sites within the HAOD range and AAOD simulated by the nearest corresponding AERONET site using WRF-Chem adding different BB inventories, where the satellite inversions of both AOD > 1 and AAOD > 0.03 range 97-110°E, 15-22.5°N are called HAOD. The legend line is the same as in Figure 7.**

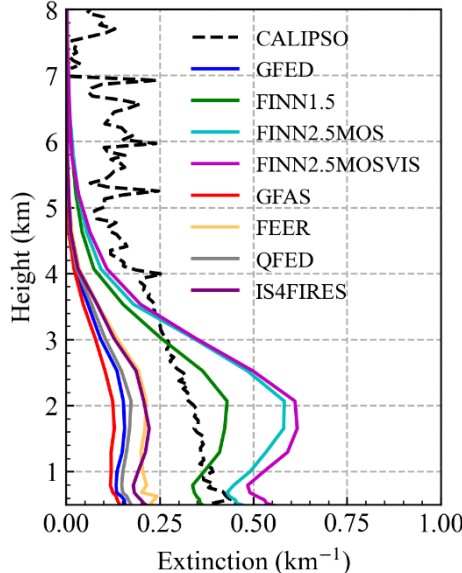

1003

**Figure 12. Vertical distributions of monthly mean aerosol extinction (550 nm) from WRF-Chem with different BB inventories and the corresponding CALIPSO retrieval (532nm) in HAOD (97-110°E, 15-22.5°N). The black dotted line indicates CLIAPSO and the remaining lines are the same as in Figure 7.**



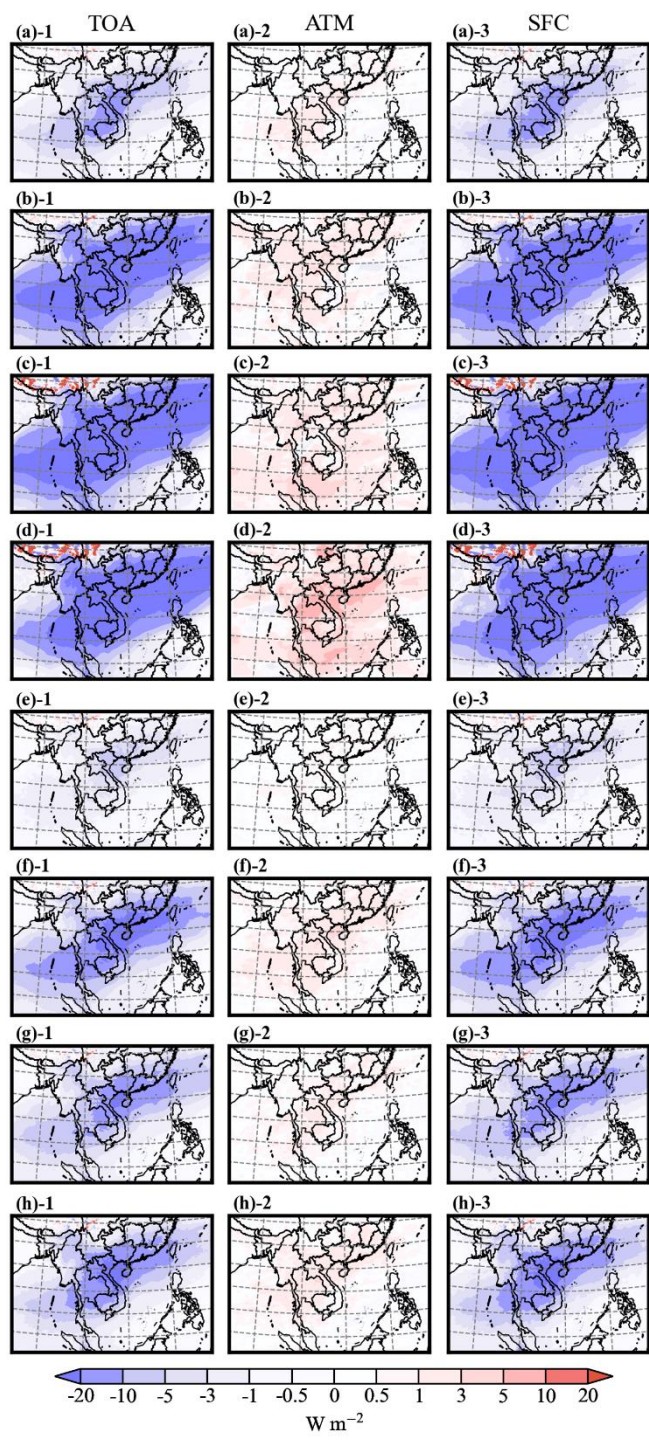

**Figure 13. The average difference in clear-sky SW radiation fluxes (daytime) simulated with and without BB emission (GFED, FINN1.5, FINN2.5 MOS, FINN2.5 MOSVIS, GFAS, FEER, QFED, IS4FIRES) over the PSEA in March 2019 at the top of the atmosphere (TOA), ground surface (SFC), and in the atmosphere (ATM), Where (a)-(h) represent 8 emission inventories.**



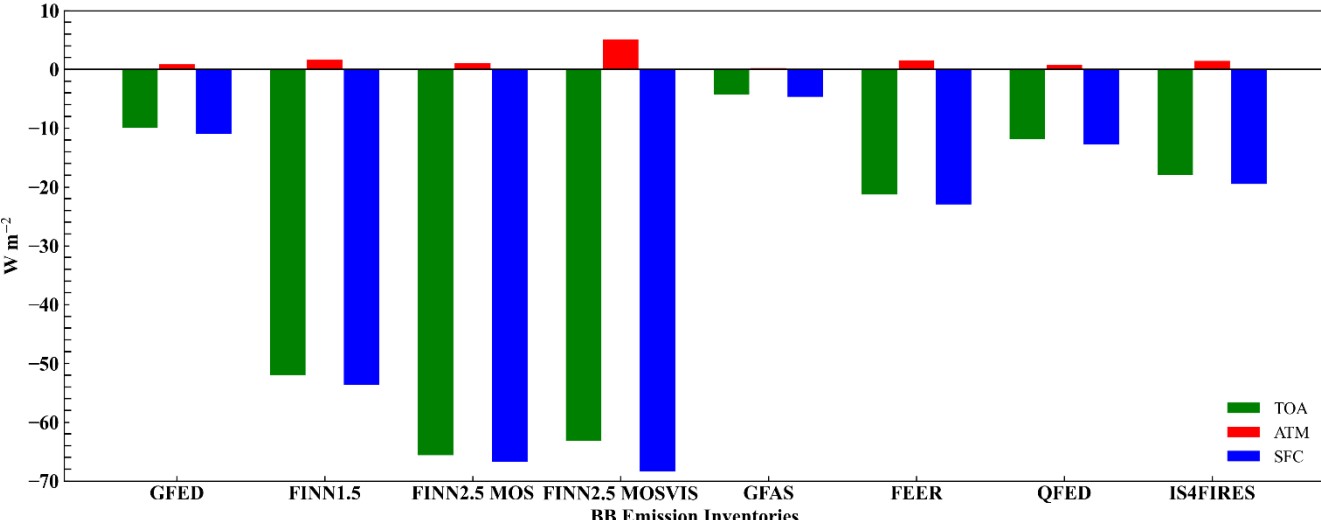

**Figure 14. The average difference in clear-sky SW radiation fluxes (daytime) simulated with and without BB emission in the HAOD (97-110°E,15-22.5°N) region during March 2019.**