# Peer review of "Measurement report: Assessing the Impacts of Emission Uncertainty"

_EGUsphere, 2023_

## Referee Comment (RC2)

Jin et al., 2023 present a comparison study of 8 biomass burning (BB) inventories using the Weather Research Forecasting model coupled with Chemistry (WRF-Chem) configured with the Model for Ozone and Related chemical Tracers (MOZART) and the Model for Simulating Aerosol Interactions and Chemistry (MOSAIC). They assess how these BB inventories impact aerosol optical properties (AOPs) such as, aerosol optical depth (AOD), aerosol absorption optical depth (AAOD), and aerosol extinction coefficients (AEC). Additionally, the direct radiative forcing (DRF) of BB aerosol was assessed. The AOPs were compared against ground and satellite-based measurements. This study is valuable to the ACP community as BB events increase in frequency, furthering the need to understand the biases certain BB inventories impose on AOPs. With that said, the authors need to address a number points prior to publication.

What is the rationale behind choosing March 2019 as the study timeframe? How does the fire season (March 2019) compare to other fire seasons in the region, was it representative of the average conditions (or anomalously high/low)?

The influence of external dust aerosol on AEC is mentioned in section 3.5, 4.2, and in the summary and conclusions. Can you provide more details on how external dust (or other inorganic aerosols e.g., sea salt aerosol) impacts the AEC profiles?

Lines 553 – 556, Jin et al. mention that when direct and indirect radiation feedbacks are included in WRF-Chem they improve the representation of AOPs, but indirect radiation feedbacks are not included in their simulations. Jin et al., mention that this, "may also lead to biases in the AOPs" (Line 556), but what specifically are those biases? Please expand on this point.

The semi-direct effect from absorbing aerosols (AAs) is another important process that impacts DRF. AAs are effective at absorbing shortwave radiation in the atmosphere and can burn-off clouds (impacting DRF). Is this process included in this modelling framework? A useful study for this may be Mallet et al., 2020.

Understanding more details of the aerosol composition in the BB inventories will be useful. How are aerosol mixing processes (external and internal mixed aerosol) included in your modelling framework? These mixing processes will impact the hygroscopicity of aerosols, impacting AOPs depending on the aerosol composition of each inventory.

Minor points are below.

Figures 3, 4, and 13 should have the inventories labelled on the top of the panel. This will make it easier to interpret the results.

Line 152 – is "gas" referring to $SO_2$ and $NH_3$ (as it is on line 310)? If so, I might suggest just stating $SO_2$ and $NH_3$ explicitly as "gas" is somewhat ambiguous.

Lines 321 – 322, Jin et al., mention that QFED exhibits a lower BC to OC ratio compared to the other inventories. Do you have any comments as to why this inventory leads to a lower BC/OC compared to the other inventories?

Line 493 – 494, "(with FINN2.5 MOSVIS reaching a maximum of 70 W m$^{-2}$)" Please make it clearer what maximum you are referring to.

In table 1, I suggest changing the "Main EF" label to "EF reference (s)". Make it clear that these are references.

(As an example) Line 45 uses "$W/m^2$", please change all instances of this to "$W\ m^{-2}$".

On figure S1, please remove the "figure" label at the top left.

Figure S2, make it clearer which letter labels refer to which of the 23 cities.

References:

Mallet, M., Solmon, F., Nabat, P., Elguindi, N., Waquet, F., Bouniol, D., Sayer, A. M., Meyer, K., Roehrig, R., Michou, M., Zuidema, P., Flamant, C., Redemann, J., and Formenti, P.: Direct and semi-direct radiative forcing of biomass-burning aerosols over the southeast Atlantic (SEA) and its sensitivity to absorbing properties: a regional climate modeling study, Atmos. Chem. Phys., 20, 13191–13216, https://doi.org/10.5194/acp-20-13191-2020, 2020.

---

## Author Comment (AC1)

We sincerely appreciate your invaluable feedback and thorough review of our research. Your insights are crucial for enhancing the quality and reliability of our study. We are highly grateful for the time and effort you've devoted to this process. Your comprehensive comments have provided us with valuable guidance, and we are committed to addressing all the issues you've raised to ensure that our research meets the highest standards. Thank you for your dedication to advancing science and your commitment to assisting us in refining our research. We look forward to resubmitting the revised manuscript and responding to your suggestions.

Main comments

**1.What is the rationale behind choosing March 2019 as the study timeframe? How does the fire season (March 2019) compare to other fire seasons in the region, was it representative of the average conditions (or anomalously high/low)?**

Response:

1. We have accepted the reviewer's suggestions and added the spatial distribution characteristics of the MODIS inversion fire points in March 2019 to Figure 1(b), as well as a histogram showing the total number of fire points for each month in 2019 in Figure 1(c). This will help to further emphasize the importance of the period used in the simulation.

2. Regarding the reviewer's inquiry about the rationale for selecting March 2019 as the study period, we have incorporated pertinent information in the manuscript.

1)Lines 83-86. " Wiedinmyer et al. (2023) have shown that the seasonal cycle (averaged over 2012-2019) of CO emissions from BB in various regions of the world and the latest version of FINN v2.5 (MODIS+ VIIRS) has an emission peak in March, primarily driven by emissions from the PSEA. However, this peak is absent in GFED and is less pronounced in other emission inventories (FINN1.5, FEER, GFAS, QFED). " Therefore, it is imperative to determine the causes of emissions from different fire sources in mainland Southeast Asia in March.

2)Lines 95-102. "The World Meteorological Organization's report highlights that the early part of 2019 corresponds to the El Niño cycle (from April to May, the temperature of waters beneath the surface of the tropical Pacific has notably declined) (Organization, 2019), during which meteorological conditions are more favourable for the occurrence and propagation of BB (Cochrane, 2009). Additionally, Yin (2020) discovered that over the past 18 years (2001-2018), the PSEA region predominantly experienced the peak of BB activity in March each year. Fan et al. (2023) and Duc et al. (2021) confirmed that the PSEA suffered severe air quality impacts during the BB in March 2019. Therefore, centered on the period of March 2019, this study aims to analyze how emission uncertainties or differences from different BB inventories affect the spatial and temporal distribution of aerosols and their radiative effects in the PSEA region."

3)Lines 111-114."Figure 1 depicts the simulation domain, outlined in blue (Figure 1(a)). It shows that the MODIS active fire instances during March 2019 were primarily consolidated in Laos, Cambodia, and Northern Thailand, as well as in Eastern and Western Myanmar (Figure 1(b)). Importantly, with a total of 69,771 fire counts, March 2019 saw the highest monthly peak of fires for that year (Figure 1(c))."

**2.The influence of external dust aerosol on AEC is mentioned in section 3.5, 4.2, and in the**

**summary and conclusions. Can you provide more details on how external dust (or other inorganic aerosols e.g., sea salt aerosol) impacts the AEC profiles?**

Response:
We accepted the reviewer's suggestion to add Figure S4 to illustrate the effect of dust and sea salt aerosols on AEC
1. Lines 486-493
"Figure S4 illustrates the frequency distribution of six aerosol types at an altitude of 8 km over the PSEA region in March 2019. Within the higher altitudes of 5-7 km the presence of dust, polluted dust, and smoke aerosols is evident, with the dust aerosols originating from the upper-level westerlies in the Indian region. Within this altitude range, the simulated AEC gradually approaches zero with increasing altitude. However, the AEC retrieved by CALIPSO exhibits three peaks, which may be attributed to uncertainties in the calculation model for BB injection heights and the influence of external dust transport."
2. Lines 580-583
"Despite the influence of sea salt aerosols in the near-surface region of PSEA (Figure S4), the contribution of sea salt aerosol to AOD is notably small, approximately 2% (Zeng et al., 2023). Additionally, Dong and Fu (2015a) observed that the model, during the period from 2006 to 2010, accurately simulated BB AOD without incorporating sea-salt emissions over the PSEA region. Consequently, our model does not consider sea-salt emission inventories."

**3.Lines 553 – 556, Jin et al. mention that when direct and indirect radiation feedbacks are included in WRF-Chem they improve the representation of AOPs, but indirect radiation feedbacks are not included in their simulations. Jin et al., mention that this, "may also lead to biases in the AOPs" (Line 556), but what specifically are those biases? Please expand on this point.**

Response:
We have made corresponding changes to lines 587-592 based on the reviewers' suggestions, which mainly involve two parts defining ACI and modeling how the absence of ACI affects aerosol optical properties.
1."Additionally, the inclusion of ARI and aerosol–cloud interactions (ACI) in the WRF-Chem model has been found to effectively improve the simulation of AOPs in European wildfire simulations, whereas this study only incorporates ARI"
2."ACI is concerned with aerosols altering albedo and lifetime of clouds (Baró et al., 2016). "
3."Failure to account for ACI may result in models that do not accurately simulate cloud droplet numbers and sizes, lifetimes, and radiative balances, with implications for climate and atmospheric AOPs (Gao et al., 2022)."

**4.The semi-direct effect from absorbing aerosols (AAs) is another important process that impacts DRF. AAs are effective at absorbing shortwave radiation in the atmosphere and can burn-off clouds (impacting DRF). Is this process included in this modelling framework? A useful study for this may be Mallet et al., 2020.**

Response:
The ARI in the WRF-Chem model includes traditional aerosol direct and semi-direct effects. We

have revised this section in the model introduction section to show that the model includes semi-direct radiative effects.

Lines 129-131

"The aerosol-radiation interactions (ARI) scheme of WRF-Chem includes the traditional aerosol direct and semi-direct effects (Baró et al., 2016). Mallet et al. (2020) and Palacios-Peña et al. (2018) found that model incorporation of ARI can effectively replicate smoke aerosol simulations, so the ARI scheme was selected for this paper."

**5. Understanding more details of the aerosol composition in the BB inventories will be useful. How are aerosol mixing processes (external and internal mixed aerosol) included in your modelling framework? These mixing processes will impact the hygroscopicity of aerosols, impacting AOPs depending on the aerosol composition of each inventory.**

Response:

We thank the reviewers for their valuable comments. All eight BB emissions use the WRF-Chem model with the MOSAIC 4 bin scheme to simulate atmospheric aerosols. In the model, we considered that the aerosol compositions were externally mixed between the size bins and internally mixed in each size bin. The internal mixing refractive index was the volume-weighted mean refractive index of each composition. This means that the different aerosol components are homogeneously mixed within the same size range, which helps to model the chemical and optical properties of the aerosol more accurately. However, the aerosol components are mixed externally between different size bins. This means that there can be different chemical compositions between aerosol particles of different size ranges, which reflects the complexity of aerosols in the actual atmosphere. This external mixing process has an important effect on the water hygroscopicity of aerosols and thus on AOPs. This study aims to analyze how the uncertainties or differences in emissions in different emission inventories in the PSEA region affect the spatial and temporal distribution characteristics of aerosols and aerosol radiative effects in the WRF-Chem model. In addition, Reddington et al. (2019) found that the modeled AOD of BB aerosols is relatively insensitive to the assumption of aerosol mixing state. Therefore, the interrelationships between different inventory emission aerosols and their modeled mixing state-absorption-optical properties are relatively less explored in this paper.

Minor points are below.

**1. Figures 3, 4, and 13 should have the inventories labelled on the top of the panel. This will make it easier to interpret the results.**

Response:

We have accepted the reviewer's comments and have carefully rechecked the figures throughout the paper, in particular adding the names of the emission inventories to Figures 3, 4, and 13 to facilitate the reader's understanding.

**2. Line 152 – is "gas" referring to SO2 and NH3 (as it is on line 310)? If so, I might suggest just stating SO2 and NH3 explicitly as "gas" is somewhat ambiguous.**

Response:

We have adjusted Line 166, where "gas" has now been changed to "gases ($CO$, $NO_X$, $SO_2$, $NH_3$)"

and the Table 1 header "Gas" has been modified to "Gases", to differentiate it from "Gas" in Line 324.

**3.Lines 321 – 322, Jin et al., mention that QFED exhibits a lower BC to OC ratio compared to the other inventories. Do you have any comments as to why this inventory leads to a lower BC/OC compared to the other inventories?**

Response:

We accept the reviewer's suggestion to make changes in lines 339-340. "In addition, differences in emission EF in Southeast Asia may result in a BC/OC equal to approximately 0.08."

**4.Line 493 – 494, "(with FINN2.5 MOSVIS reaching a maximum of 70 W m-2)" Please make it clearer what maximum you are referring to.**

Response:

It refers to the DRF of the FINN25 MOSVIS scheme in Figure 10 that simulates SFCs in the PESA region with a maximum of about 70 W m$^{-2}$. We accept the reviewer's suggestion to make changes at lines 516-517."The eight schemes simulate the DRF of -32.60±24.50 W m$^{-2}$ at SFC in the daytime with FINN2.5 MOSVIS reaching a maximum of approximately70 W m$^{-2}$ (Figure 10)"

**5.In table 1, I suggest changing the "Main EF" label to "EF reference (s)". Make it clear that these are references.**

Response:

We accepted the reviewers' comments and revised Table 1

**6.(As an example) Line 45 uses "W/m2", please change all instances of this to "W m-2".**

Response:

We thank the reviewers for their comments, and we have revised the manuscript in its entirety in accordance with the journal's requirements.

**7.On figure S1, please remove the "figure" label at the top left.**

Response:

The "figure" has been removed.

**8.Figure S2, make it clearer which letter labels refer to which of the 23 cities.**

Response:

Thank you for the reviewer's suggestions. In order to better illustrate the comparison between simulated and observed data for various city sites, we have added city titles to Figure S2.

Reference

Reddington, C. L., Morgan, W. T., Darbyshire, E., Brito, J., Coe, H., Artaxo, P., Scott, C. E., Marsham, J., and Spracklen, D. V.: Biomass burning aerosol over the Amazon: analysis of aircraft, surface and satellite observations using a global aerosol model, Atmos. Chem. Phys., 19, 9125-9152, 10.5194/acp-19-9125-2019, 2019.

---

## Author Comment (AC2)

We sincerely appreciate the detailed feedback you provided on our manuscript, which greatly assisted us in our research endeavors. We have addressed the rationale for choosing March 2019 as the study period by adding additional information, including citations and the data in Figure 1, to better emphasize the importance of this choice. We have explained in detail that the BB emissions inventory provides aerosol types and how the model handles inorganic aerosols to satisfy your questions about aerosols. In addition, we add in the discussion section that insufficient inversion of the BB emissions inventory can lead to uncertainties. Regarding the comparison with satellite products, we clarified that the purpose of this comparison is to help evaluate the accuracy and performance of the model, especially under different BB emission inventories. At the same time, we emphasize our main focus on the differences in aerosol mass concentrations and their spatial and temporal distributions in the PSEA region for the different emission inventories, as well as the use of the WRF-Chem model with the same model configuration to assess aerosol optical properties and radiative forcing during BB in the PSEA region. A detailed line-by-line response is provided below.

Main comments

**(1)To further emphasize why the month that was used for the simulations as it was one line in the introduction that may be lost, it would be good in Fig. 1 (or another figure), to show the total fire counts in Peninsular Southeast Asia.**

Response:

1. We have accepted the reviewer's suggestions and added the spatial distribution characteristics of the MODIS inversion fire points in March 2019 to Figure 1(b), as well as a histogram showing the total number of fire points for each month in 2019 in Figure 1(c). This will help to further emphasize the importance of the period used in the simulation.

2. Regarding the reviewer's inquiry about the rationale for selecting March 2019 as the study period, we have incorporated pertinent information in the manuscript.

1)Lines 83-86. " Wiedinmyer et al. (2023) have shown that the seasonal cycle (averaged over 2012-2019) of CO emissions from BB in various regions of the world and the latest version of FINN v2.5 (MODIS+ VIIRS) has an emission peak in March, primarily driven by emissions from the PSEA. However, this peak is absent in GFED and is less pronounced in other emission inventories (FINN1.5, FEER, GFAS, QFED) " Therefore, it is imperative to determine the causes of emissions from different fire sources in mainland Southeast Asia in March.

2)Lines 95-103.  "The World Meteorological Organization's report highlights that the early part of 2019 corresponds to the El Niño cycle (from April to May, the temperature of waters beneath the surface of the tropical Pacific has notably declined) (Organization, 2019), during which meteorological conditions are more favourable for the occurrence and propagation of BB (Cochrane, 2009). Additionally, Yin (2020) discovered that over the past 18 years (2001-2018), the PSEA region predominantly experienced the peak of BB activity in March each year. Fan et al. (2023) and Duc et al. (2021) confirmed that the PSEA suffered severe air quality impacts during the BB in March 2019. Therefore, centered on the period of March 2019, this study aims to analyze how emission uncertainties or differences from different BB inventories affect the spatial and temporal distribution of aerosols and their radiative effects in the PSEA region."

**(2)How does the model treat the inorganic aerosol? E.g., it is not clear if the inorganic aerosol**

**is treated thermodynamically or not. This is important to better understand how the model may be treating aerosol liquid water, aerosol acidity, etc., which all impact the physicochemical properties of the aerosol and thus the aerosols' optical properties.**

Response:

In this study, we use the WRF-Chem model to treat inorganic aerosols through the Model for Simulating Aerosol Interactions and Chemistry (MOSAIC) mechanism. The MOSAIC mechanism has been designed with a highly modular structure to facilitate seamless coupling between various chemical and microphysical processes. The current version of the model incorporates several key features and modules, including the treatment of inorganic aerosols. The following are the key aspects related to the treatment of inorganic aerosols in the model:

1. MOSAIC explicitly addresses various important inorganic aerosol species that are significant at urban, regional, and global scales (Zaveri et al., 2008). These species encompass sulfate, methanesulfonate, nitrate, chloride, carbonate, ammonium, sodium, calcium, black carbon (BC), primary organic mass (OC), and liquid water. Unspecified inorganic species such as silica, other inert minerals, and trace metals are grouped together as 'other inorganic mass' (OIN). The model also accounts for the gas-phase species that can partition to the particle phase, which includes $H_2SO_4$, $HNO_3$, HCl, $NH_3$, and MSA (methanesulfonic acid). Work is ongoing to include the treatment of secondary organic aerosols (SOA).

2. The MOSAIC model incorporates a thermodynamic module that enables the accurate prediction of particle deliquescence, water content, and solid-liquid phase equilibrium in multicomponent aerosols at specific relative humidity (RH) and temperature (T) conditions. This module is crucial for computing the mass transfer driving forces for dynamic gas-particle partitioning of various semivolatile species. The thermodynamic module is specially designed to be both accurate and computationally efficient for 3-D modeling applications.

In summary, the MOSAIC mechanism in the WRF-Chem model treats inorganic aerosols through a thermodynamic approach, explicitly considering several important inorganic species. It also accurately accounts for phase equilibria and deliquescence, which are crucial for understanding aerosol liquid water content, aerosol acidity, and their impact on aerosol physicochemical and optical properties. The comprehensive treatment of inorganic aerosols in the model ensures its reliability and applicability across various scales and applications. Many scholars have already used this mechanism to simulate aerosol optical properties (AOPs) during the BB period, and they have obtained reliable research results (Palacios-Peña et al., 2018; Archer-Nicholls et al., 2015; Wu et al., 2017). Therefore, modifications were made in the manuscript from line 127 to line 128 (other inorganic aerosols through a thermodynamic approach, with high efficiency and accuracy for use in air quality and regional/global aerosol modeling (Zhang et al., 2018).).

**(3) As the results are presented, it is currently not clear what the purpose of the satellite products comparisons with the model results provides for the conclusions. E.g., the authors discuss how different emission inventories provide different agreement depending on the satellite product and/or land-based product, which indicates no emission inventory is superior. Further, the authors have not provided or discussed the following properties that would be**

**potentially of more interest/importance in understanding the aerosol from biomass burning to compare with observations and products:**

Response:

1. This study aims to analyze the differences in mass concentration and spatial-temporal distribution of pollutants in different BB emission inventories, and how their incorporation into WRF-Chem affects regional air quality, the spatiotemporal distribution of aerosols, and aerosol radiative effects in the PSEA region. Our focus is not on analyzing the micro-level differences in BB emission inventories from the bottom up (such as classification of organic and inorganic aerosols, aerosol size, oxidation, etc.) and selecting aerosol mechanisms for models. Instead, we have compiled and analyzed eight BB emission inventories to provide the necessary scientific basis for other scholars to simulate AOPs and radiation forcing in the PSEA region based on BB emission inventories. Specifically, these emission inventories share identical model configurations, including meteorological initial and boundary conditions, gas-phase chemistry, and aerosol mechanisms, as well as the same geographical region and study period. Through this design, our research contributes to revealing differences and uncertainties among different BB emission inventories, particularly concerning the PSEA region. We aim to understand how different inventories capture BB emissions and their effects on aerosol and gas emissions. Additionally, the paper also focuses on AOPs, such as aerosol optical depth (AOD), absorbing aerosol optical depth (AAOD), and aerosol extinction coefficient (AEC). Furthermore, it analyzes the impact of different BB emission inventories on direct radiative forcing (DRF), aiming to assess their impact on atmospheric radiation balance and better understand the impact of aerosols on climate. Our results suggest that FINN1.5 and IS4FIRES are recommended for accurately assessing the impact of BB on air quality and climate in the PSEA region.

2. Satellite remote sensing products can provide reliable observational data over a large area compared to ground stations, which has advantages for studying the large-scale impact of BB. In addition, the products are not affected by human disturbances, making them more representative of the confirmed environment of atmospheric aerosols. Many scholars have used satellite remote sensing products to evaluate model simulations of aerosol optical characteristics during BB (Palacios-Peña et al., 2018; Reddington et al., 2016). This paper first validates satellite remote sensing products with ground stations before using them as evaluation data for models, so the data is reliable. By comparing the model-simulated aerosol optical characteristics with satellite products, researchers can evaluate the accuracy and performance of the model. This helps to determine the reliability and usability of the model, especially when simulating atmospheric aerosols. Such evaluation helps to identify the limitations and room for improvement of the model.

**(a) What is the aerosol composition with each emission inventory? E.g., how much primary vs secondary organic carbon/aerosol? How much secondary inorganic aerosol vs organic aerosol? How much black carbon vs these other components? All these aspects impact the hygroscopicity of the aerosol, and thus how it would be retrieved by satellite and ground-based measurements.**

Table 1 presents the different aerosol types in the emission inventories. Currently, the emission factors used in the calculation of BB emission inventories are derived based on smoke samples

collected at low altitudes (sampled after any rapid initial cooling but before most photochemical reactions). The aerosols retrieved from these samples only provide information on particulate matter concentrations and do not include classified products for different aerosol types (e.g., secondary inorganic aerosol). Andreae and Merlet (2001) indicate that obtaining aerosol concentrations from emission factors should be regarded as rather crude estimates. They are intended for application to lightly aged plumes (1-2 hours) to avoid significant temporal changes shortly after emissions. Results for different particle categories seem to be quite consistent internally, even when they originate from various sources. Therefore, the BB emission inventories selected in this study (except for IS4FIRES, which only provides $PM_{2.5}$ mass concentration) provide BC, OC, $PM_{2.5}$, or $PM_{10}$ mass concentrations and do not include the classification of secondary organic aerosols and secondary inorganic aerosols.

Figure 4 illustrates the species distribution in each emission inventory, including BC. Although the varying proportions of aerosol types in BB emissions do not constitute the main focus of this paper regarding aerosol hygroscopicity, we have included this aspect in the discussion section (Lines 558-561: Furthermore, the representation of aerosols in the BB emission inventories is insufficient, including chemical components, size distribution of aerosols, aging processes, hygroscopic growth, vertical and horizontal transport (including the injection height of fire emissions), and oxidation state (Reddington et al., 2016), which can all lead to modeling biases in AOPs. Importantly, these attributes also have an impact on aerosols in cloud and radiative forcing.).

**(b) How does the size distribution change amongst the different emission inventories? Similar to the chemical composition, the sizes would impact both water uptake, scattering, and how well the satellite and ground-based observations detect the aerosol.**

The size distribution of BB aerosols ranges from tens of nanometers to millimeters in a continuous spectrum, with the majority of the mass existing in the mode of several hundred nanometers (Reid et al., 2005). Mass concentration measurements are typically reported as $PM_1$, $PM_{2.5}$, $PM_{10}$, or TPM, representing size ranges less than 1, 2.5, and 10 µm, as well as the total particulate matter mass. In this study, the construction of aerosol emission factors for the eight emission inventories is derived from Andreae (2019); Akagi et al. (2011); Andreae and Merlet (2001), with $PM_{2.5}$, $PM_1$–$PM_5$ categorized as $PM_{2.5}$, and $PM_{10}$ representing the $PM_{10-2.5}$ fraction. Differences in aerosols within BB emission inventories can impact water absorption, scattering, and the ability of satellites and ground-based observations to detect aerosols. Therefore, we have included this aspect in the discussion section (Lines 558-562).

**(c) What is the oxidation state, e.g., O/C and H/C ratio, of the primary and secondary aerosol? Similar to (a), the amount of oxidation of the organic aerosol/carbon will impact its physicochemical properties and how it would be retrieved.**

As mentioned in the response in (a), the BB emission inventories studied in this paper only derive BC, OC, $PM_{2.5}$, or $PM_{10}$ mass concentrations in an idealized manner, based on in-situ measurements of young fire plumes. They do not include information on primary aerosols and the oxidative state of secondary aerosols. We have incorporated this aspect into the discussion section of the article

(Lines 558-562).

**(d) Besides retrieval, all these properties would impact the aerosols role in clouds and radiative forcing, making it important to understand how much these differences may impact the differences presented in the different figures.**

1. In our model, only the influence of ARI was considered, and the impact of ACI was not taken into account. We have added a section in the Discussion (Lines 587-592: Additionally, the inclusion of ARI and aerosol–cloud interactions (ACI) the inclusion of direct and indirect radiation feedback in the WRF-Chem model has been found to effectively improve the simulation of AOPs in European wildfire simulations (Palacios-Peña et al., 2019), whereas this study only incorporates ARI. ACI is concerned with aerosols altering the albedo and lifetime of clouds (Baró et al., 2016). Failure to account for ACI may result in models that do not accurately simulate cloud droplet numbers and sizes, lifetimes, and radiative balances, with implications for climate and atmospheric AOPs (Gao et al., 2022).).

2. The primary focus of this paper is to investigate the differences in aerosol mass concentrations and their spatiotemporal distribution among various BB emission inventories, as well as to assess AOPs and radiative forcing during BB events in the PSEA region using the WRF-Chem model with identical configurations. The reviewer has rightly pointed out that the classification of chemical species (primary organic, secondary inorganic, etc.), size distribution, oxidative characteristics, hygroscopic growth of BB aerosols in the emission inventories can significantly impact the simulation of AOPs and aerosol-cloud-radiation interactions, which are indeed crucial research topics. However, these aspects are not the primary focus of this paper and were not extensively explored. We have included this discussion in the article's discussion section (Lines 558-562).

**(3) Without the information provided in (2), the intercomparisons of the model and observed PM2.5 is hard to interpret, as the models may be getting PM2.5 correct for the incorrect reason. Also, it is unclear in the intercomparison of the model with observed PM2.5 for one fire emission inventory how to interpret the results as (a) it seems most of the PM2.5 was measured in urban areas, meaning the urban emissions may be driving the intercomparison more than fire emissions and (b) the emission inventory used for the intercomparison and validation of the model has mixed results (e.g., Table 2).**
Response:
1. We have provided additional information regarding point (2) and elaborated on it in the discussion section (Lines 558-561). While these characteristics can impact the inversion of BB emission inventories and subsequently influence the simulation results, the primary focus of this paper is the investigation of differences in aerosol mass concentrations and their spatiotemporal distribution among different BB emission inventories. We aim to assess their impact on the simulation of AOPs and radiative forcing. Our modeling approach maintains identical meteorological initial and boundary conditions, gas-phase chemistry, aerosol mechanisms, and covers the same geographical region and study period. Hence, we are capable of exploring how differences in aerosol mass concentrations and their spatiotemporal distribution in BB emission inventories affect the simulation of $PM_{2.5}$.

2. Comparisons of our simulated $PM_{2.5}$ with data from 23 monitoring stations indicate that the model is capable of reasonably reproducing the spatiotemporal distribution characteristics of pollutants (Figure S2). Notably, several stations in high BB emission areas, such as Chiang Rai Mueang in northern Thailand and Jinghong in China, show better performance (Table S7, with R values of 0.64 and 0.75, respectively) compared to stations located farther away from high BB emission areas. Furthermore, the results of all our stations compared with observations show better simulation performance in this region during BB events, in contrast to previous studies by other scholars (Lines 355-359).

3. Table 2 shows the comparison of the AOD and AAOD simulated by the model with the addition of the eight BB emission inventories with the AOD from the MODIS inversion and the AAOD from the TROPOMI inversion in the BB high-emission areas. The aerosol concentration in the FINNs emission inventory is significantly higher than the other emissions, so the simulation results show an overestimation and the others an underestimation. The AAOD also shows this trend, which was also found in Zhang et al. (2014). In addition, the difference in aerosol concentrations among the eight emission inventories was 11 times, but the simulated AOD and AAOD differences were reduced. These smaller differences in modeled variables may reflect atmospheric dispersion and deposition effects.

**(4) Due to (2) and (3), the paper may be presented better as a comparison against the emission inventories without comparison with satellite and ground based products as it is not clear that there is a better emission inventory to used currently for chemical transport models. More discussion could be placed into the description in the similarity and differences in the physicochemical properties due to differences in the emission inventory, which would be of extreme interest towards the community.**

Response:

1. Although the current BB aerosol inventories inverted by remotely sensed satellites or ground-based observations are somewhat deterministic due to (2) and (3), BB emission inventories are still able to characterize the spatial and temporal distribution of pollutants in large-scale BB emissions, and a large number of scholars have also analyzed BB events through the use of these emission inventories in models (Reddington et al., 2016; Zhang et al., 2014).

2. With the increasing frequency of global fires due to global warming, more and more scholars have studied the characteristics of wildfires through numerical simulations, in which the necessary input data for the model is the BB emission inventory. A large number of scholars have evaluated the applicability of BB emission inventories by comparing the performance of multiple emission inventories in models, and have given the most applicable emission inventories for global simulations or regional simulations (Desservettaz et al., 2022; Pan et al., 2020). Our results show that in the PSEA region, FINN1.5 and IS4FIRES schemes are recommended.

3. We greatly appreciate your interest in the physicochemical properties of BB aerosols in retrieval and modeling. In the paper, we have included relevant discussions (Lines 558-572).

Minor

**(1) For all figures, please label either which emission inventory is being used or what location the observations/model is for. It is currently difficult to interpret the figures without this key**

**information.**

Response:

We have accepted the reviewer's comments and have carefully rechecked the figures throughout the paper, in particular adding the names of the emission inventories to Figures 3, 4, and 13 to facilitate the reader's understanding.

**(2) Please check figures and tables. There are many instances of inconsistencies or typos in the labels (e.g., line 103 says red line around the study area, Fig. 4 has methanal which is formaldehyde and then an abbreviation for methylglyoxal (Mgly) and methyl vinyl ketone twice with MACR for one, etc).**

Response:

Thank you for your critical feedback and attention to detail. We deeply appreciate your effort in identifying these inconsistencies and typos in the figures and tables. We have since duly revised Lines 111-116 to accurately reflect the study area demarcation in the referenced figure. Additionally, we have corrected the labeling errors in Figure 4 ("methylglyoxal (MGLY), glyoxal (GLY), methacrolein (MACR), and lumped monoterpenes, as α-pinenedecane ($C_{10}H_{16}$)").

**(3) It is highly recommended to not use rainbow for color bars. Rainbow color bars can be difficult to interpret due to color blindness and the contrast between colors can be difficult to observe differences. Similarly, the color bar in Fig. 6c and Fig. 10c is extremely difficult to read and interpret any differences.**

Response:

We have considered the issues raised by the reviewers, particularly those related to the rainbow color spectrum and the color bars in the figure. We have modified the colors in Figures 6c and 10c to improve the readability and interpretability of the figures.

**(4) Table S4. Please include location for each met station.**

Response:

We have accepted the reviewer's feedback and supplemented the latitude and longitude data in Table S6.

**(5) Please introduce the supplemental figures and tables in numerical order. E.g., right now, one supplemental table with a higher numerical value is introduced prior to a lower numerical value table, making the reader jump between tables.**

Response:

We sincerely appreciate the reviewer's suggestions. We have re-checked the order of figures and tables in the supplements as they appear in the manuscript and conducted the following analysis. We found that the current order of citation is correct.

| Figures | The first occurrence of line number |
|---------|-------------------------------------|
| Figure S1 | 314 |
| Figure S2 | 351 |
| Figure S3 | 365 |
| Figure S4 | 486 |

| Tables | The first occurrence of line number |
|---|---|
| Table S1 | 180 |
| Table S2 | 197 |
| Table S3 | 197 |
| Table S4 | 205 |
| Table S5 | 216 |
| Table S6 | 270 |
| Table S7 | 273 |
| Table S8 | 298 |
| Table S9 | 461 |

Reference

Akagi, S. K., Yokelson, R. J., Wiedinmyer, C., Alvarado, M. J., Reid, J. S., Karl, T., Crounse, J. D., and Wennberg, P. O.: Emission factors for open and domestic biomass burning for use in atmospheric models, Atmos. Chem. Phys., 11, 4039-4072, 10.5194/acp-11-4039-2011, 2011.

Andreae, M. O.: Emission of trace gases and aerosols from biomass burning – an updated assessment, Atmos. Chem. Phys., 19, 8523-8546, 10.5194/acp-19-8523-2019, 2019.

Andreae, M. O. and Merlet, P.: Emission of trace gases and aerosols from biomass burning, 15, 955-966, https://doi.org/10.1029/2000GB001382, 2001.

Archer-Nicholls, S., Lowe, D., Darbyshire, E., Morgan, W. T., Bela, M. M., Pereira, G., Trembath, J., Kaiser, J. W., Longo, K. M., Freitas, S. R., Coe, H., and McFiggans, G.: Characterising Brazilian biomass burning emissions using WRF-Chem with MOSAIC sectional aerosol, Geosci. Model Dev., 8, 549-577, 10.5194/gmd-8-549-2015, 2015.

Desservettaz, M. J., Fisher, J. A., Luhar, A. K., Woodhouse, M. T., Bukosa, B., Buchholz, R. R., Wiedinmyer, C., Griffith, D. W. T., Krummel, P. B., Jones, N. B., Deutscher, N. M., and Greenslade, J. W.: Australian Fire Emissions of Carbon Monoxide Estimated by Global Biomass Burning Inventories: Variability and Observational Constraints, 127, e2021JD035925, https://doi.org/10.1029/2021JD035925, 2022.

Palacios-Peña, L., Baró, R., Baklanov, A., Balzarini, A., Brunner, D., Forkel, R., Hirtl, M., Honzak, L., López-Romero, J. M., Montávez, J. P. J. A. C., and Physics: An assessment of aerosol optical properties from remote-sensing observations and regional chemistry–climate coupled models over Europe, 18, 5021-5043, 2018.

Pan, X., Ichoku, C., Chin, M., Bian, H., Darmenov, A., Colarco, P., Ellison, L., Kucsera, T., da Silva, A., Wang, J., Oda, T., and Cui, G.: Six global biomass burning emission datasets: intercomparison and application in one global aerosol model, Atmos. Chem. Phys., 20, 969-994, 10.5194/acp-20-969-2020, 2020.

Reddington, C. L., Spracklen, D. V., Artaxo, P., Ridley, D. A., Rizzo, L. V., and Arana, A.: Analysis of particulate emissions from tropical biomass burning using a global aerosol model and long-term surface observations, Atmos. Chem. Phys., 16, 11083-11106, 10.5194/acp-16-11083-2016, 2016.

Reid, J. S., Koppmann, R., Eck, T. F., and Eleuterio, D. P.: A review of biomass burning emissions part II: intensive physical properties of biomass burning particles, Atmos. Chem. Phys., 5, 799-825, 10.5194/acp-5-799-2005, 2005.

Wu, Y., Han, Y., Voulgarakis, A., Wang, T., Li, M., Wang, Y., Xie, M., Zhuang, B., and Li, S.: An agricultural biomass burning episode in eastern China: Transport, optical properties, and impacts on regional air quality, 122, 2304-2324, https://doi.org/10.1002/2016JD025319, 2017.

Zaveri, R. A., Easter, R. C., Fast, J. D., and Peters, L. K.: Model for Simulating Aerosol Interactions and Chemistry (MOSAIC), 113, 10.1029/2007jd008782, 2008.

Zhang, F., Wang, J., Ichoku, C., Hyer, E. J., Yang, Z., Ge, C., Su, S., Zhang, X., Kondragunta, S., Kaiser, J. W., Wiedinmyer, C., and da Silva, A.: Sensitivity of mesoscale modeling of smoke direct radiative effect to the emission inventory: a case study in northern sub-Saharan African region, Environmental Research Letters, 9, 075002, 10.1088/1748-9326/9/7/075002, 2014.

---

## Author Response (AR2)

Response to Technical Corrections Request for Manuscript Submission

Dear editors and reviewers thank you very much for your valuable comments on our work, which have been very helpful in improving and enhancing our research. We appreciate the constructive feedback and understand the importance of ensuring our maps and charts are accessible to all readers, including those with color vision deficiencies. Following your suggestion, we have utilized the Coblis – Color Blindness Simulator to review our figures. We have revised the color schemes in our maps and charts to enhance readability and ensure accurate interpretation by readers with various forms of color vision deficiencies. These modifications have been made with careful consideration to maintain the integrity and clarity of the data presented. We believe these changes will make our findings more inclusive and accessible to a broader audience.

The revised figures, along with a brief description of the changes made, have been included in the updated manuscript. We hope that these adjustments meet the technical requirements for final publication. A detailed line-by-line response is provided below.

1. The title of Figure 1 has been updated to include "topography based on the digital elevation model (DEM)," and Figure 1(b) has been revised in colormap and colorbar range to illustrate the spatial distribution of fire spots in the PSEA region in March 2019.

2. Figure 3 has been modified in the colorbar range to highlight the data features.

3. Figures 6(a) and (b) have been altered in colormap, with (b) specifically revised in the colorbar range to both display data features and aid visually impaired readers in smooth reading.

4. Figures 10(a) and (b) have undergone changes in colormap, especially (b), which has an adjusted colorbar range to display data features and facilitate reading for visually impaired readers.

5. Figure 13 has been updated in both colormap and colorbar range, aiming to display data features and assist visually impaired readers.

6. Figure 14 has undergone modifications in the shape of the bar graph to enhance readability for visually impaired readers.

7. Figure S1 has been revised, with water bodies colored blue.

8. The red font in Tables S1 and S4 has been changed to black.